

SciPost Phys. Lect. Notes 86 (2024)

# An introduction to infinite projected entangled-pair state methods for variational ground state simulations using automatic differentiation

Jan Naumann[1⋆∘], Erik Lennart Weerda[2†∘], Matteo Rizzi[2,3],
Jens Eisert[1,4] and Philipp Schmoll[1‡]

**1** Dahlem Center for Complex Quantum Systems and Institut für Theoretische Physik,
Freie Universität Berlin, Arnimallee 14, 14195 Berlin, Germany
**2** Institute for Theoretical Physics, University of Cologne, 50937 Köln, Germany
**3** Forschungszentrum Jülich GmbH, Institute of Quantum Control,
Peter Grünberg Institut (PGI-8), 52425 Jülich, Germany
**4** Helmholtz-Zentrum Berlin für Materialien und Energie,
Hahn-Meitner-Platz 1, 14109 Berlin, Germany

⋆ j.naumann@fu-berlin.de , † weerda@thp.uni-koeln.de , ‡ philipp.schmoll@fu-berlin.de

## Abstract

Tensor networks capture large classes of ground states of phases of quantum matter faithfully and efficiently. Their manipulation and contraction has remained a challenge over the years, however. For most of the history, ground state simulations of two-dimensional quantum lattice systems using (infinite) projected entangled pair states have relied on what is called a time-evolving block decimation. In recent years, multiple proposals for the variational optimization of the quantum state have been put forward, overcoming accuracy and convergence problems of previously known methods. The incorporation of automatic differentiation in tensor networks algorithms has ultimately enabled a new, flexible way for variational simulation of ground states and excited states. In this work we review the state-of-the-art of the variational iPEPS framework, providing a detailed introduction to automatic differentiation, a description of a general foundation into which various two-dimensional lattices can be conveniently incorporated, and demonstrative benchmarking results.

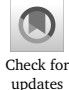

## Contents

---

∘ These authors contributed equally to the development of this work.

# 1 Introduction

Tensor networks are at the basis of a wealth of methods that are able to efficiently capture systems with many degrees of freedom, primarily in the context of interacting quantum systems, but also in a wide range of other fields. They have a long history: The beginnings can be seen [1] as originating from work on transfer matrices [2] for two-dimensional classical Ising models and methods of corner transfer matrices again in the context of classical spin models [3]. In more recent times, the rise of tensor networks to describe interacting quantum many-body systems can be traced back to at least two strands of research. On the one hand, the now famous *density matrix renormalization group* (DMRG) approach [4,5] can be regarded as a variational principle over *matrix product states* [6–8], a particularly common class of one-dimensional tensor network states. What are called *finitely-correlated states* [9] have later been understood as a Heisenberg picture variant of essentially the same family of states. These families of quantum states could further be interpreted as basically parametrizing gapped phases of matter in one spatial dimension. In a separate development, *tensor trains* became a useful tool in numerical mathematics [10]. These strands of research had been developing independently for quite a while before being unified in a common language of *tensor networks* (TN) as it stands now as a pillar of research on numerical and mathematical quantum many-body physics [11–15].

Two-dimensional tensor networks, now known as *projected entangled pair states* [16], again have a long history. The intuition why they provide a good Ansatz class for describing ground states of gapped quantum many-body Hamiltonians [17,18] – as well as other families of states – is the same as for matrix product states: Such states are expected to be part of what is called the *"physical corner"* of the Hilbert space. These states feature local entanglement compared to the degrees of entanglement unstructured states would exhibit. Ground states of gapped phases of matter are thought to satisfy *area laws for the entanglement entropy* [15]. Even though some of the rigorous underpinning of this mindset is less developed in two spatial dimensions compared to the situation in one spatial dimension, there is solid evidence that projected entangled pair states provide an extraordinarily good and powerful Ansatz class for meaningful states of two-dimensional quantum systems.

There is a new challenge arising in such two-dimensional tensor networks. In contrast to matrix product states, they cannot be exactly efficiently *contracted*: On general grounds, there are complexity theoretic obstructions against the efficient contraction of projected entangled pair states in worst case [19] – and even in average case [20] – complexity. The burden can be lessened by acknowledging that projected entangled pair states can be contracted in quasi-polynomial time [21]. These more conceptual insights constitute an underpinning of a quite practically minded question: This shows that to develop ways of efficiently and feasibly approximating tensor network contractions in two spatial dimensions is at the heart of the method development in the field.

Consequently, over the years, several numerical methods of approximately contracting projected entangled pair states have been developed. In fact, much of the method development has been along these lines. In the focus of attention in this work are projected entangled pair states directly in the thermodynamic limit, commonly referred to as *infinite projected entangled pair states* (iPEPS) [22–24]. The contraction necessary to compute expectation values of local observables gives rise to the challenge of approximately calculating effective environments. Over the years, several methods have been introduced and pursued, including methods based on boundary matrix product states [22], corner transfer matrix methods [24–26] – particularly important for the method development presented here – and tensor coarse-graining techniques [27–30].

Variational optimization algorithms for uniform matrix product states have been developed that combine density matrix renormalization group methods with matrix product state tangent space concepts to find ground states of one dimensional quantum lattices in the thermodynamic limit [31,32], building on earlier steps of devising geometrically motivated variational principles for tensor network states [33,34]. The pursuit of such variational optimization has been particularly fruitful in the two dimensional case of iPEPS. Initially proposed methods constructed the gradient of the energy explicitly using specialized environments [35,36].

Recently, as an element of major method development, the programming technique called *automatic differentiation*, widely used in the machine learning community, has been utilized for the task of calculating the gradient [37] in tensor network optimization. This step drastically simplifies the programming involved and allows one to use variational ground state search on, e.g., more exotic lattice geometries with little additional effort. Such variational approaches for iPEPS constitute the basis for this work. Automatic differentiation has also been employed in further fashions in the tensor network context in several works recently [38–41, 41–49], some of which are accompanied by publicly available code libraries [50–53]. Notably, even for gapped local Hamiltonians with chiral topological ground states, for which the numerical applicability of PEPS was unclear due to no-go theorems in related cases [54], the use of variational optimization has proven successful [41,49,55]. As a novel programming paradigm, automatic differentiation composes parameterized algorithmic components in such a way that the program becomes differentiable and its components can be optimized using gradient search. It is a sophisticated way to evaluate the derivative of a function specified by a computer program, specifically by applying the chain rule to elementary arithmetic operations. Again, it has only recently been appreciated how extremely powerful such tools are in the study of interacting quantum matter by means of tensor networks.

In this review article, we elaborate on these developments and comprehensively present ideas for a variational iPEPS method based on automatic differentiation. This includes a detailed description of the methodology and practical insights for implementations, complementing and extending the existing body of literature.

We further introduce a versatile framework, that allows arbitrary unit cells and different two-dimensional lattices to be treated on a common footing.

At the same time, this work accompanies the publicly available numerical library *variPEPS* – a versatile tensor network library for variational ground state simulations in two spatial dimensions – which implements the methods described in this review [56–58].

The content of this work is organised in three main sections. In Sec. 2, we describe the central methods that are being used in the variational iPEPS framework as well as practical remarks regarding implementation. Furthermore, we explain in detail the basics of automatic differentiation and its application in state-of-the-art ground-state search.

In Sec. 3, we then turn to explaining how to conveniently map generic lattice structures to a square one, over which the variational iPEPS methods naturally operate. Following up on this, in Sec. 4, we present numerical benchmarks obtained with the methods outlined in the previous sections and implemented in the *variPEPS* library, in comparison to other customary methods like exact diagonalization, iPEPS imaginary-time evolution and variational Monte Carlo methods.

## 2 Variational iPEPS

We seek to find the TN representation of the state vector $|\psi\rangle_{\text{TN}}$ that best approximates the true ground state vector $|\psi_0\rangle$ of an Hamilton operator of the form

$$H = \sum_{j \in \Lambda} T_j(h), \tag{1}$$

where $T_j$ is the translation operator on the lattice $\Lambda$, and $h$ is a generic *k-local Hamiltonian*, i.e., it includes an arbitrary number of operators acting on lattice sites at most at a (lattice) distance $k$ from a reference lattice point. Such a situation is very common in condensed matter physics, to say the least. To this aim, we employ the variational principle

$$\frac{\langle\psi|H|\psi\rangle}{\langle\psi|\psi\rangle} \geq E_0, \qquad \forall\, |\psi\rangle, \tag{2}$$

and use an energy gradient with respect to the tensor coefficients to search for the minimum – the precise optimization strategy being discussed later. Such an energy gradient is accessed by means of tools from *automatic differentiation* (AD), a set of techniques to evaluate the derivative of a function specified by a computer program that will be summarized below. Since we directly target systems in the thermodynamic limit, a *corner transfer matrix renormalization group* (CTMRG) procedure constitutes the backbone of the algorithm, and also will come in handy for AD purposes. This is used to compute the approximate contraction of the infinite lattice, which is crucial in order to compute accurate expectation values in the first place. Importantly, the CTMRG routine is *always* performed on a regular square lattice, for which it can be conveniently defined. Support for other lattices, also non-bipartite ones, is possible by different lattice mappings, as we will demonstrate.

Besides the CTMRG procedure, other well-controlled numerical methods have been developed to contract the infinite PEPS network. These include boundary matrix product states [22, 59] and tensor coarse-graining techniques [27–30]. These methods can be equally successfully combined with the concept of automatic differentiation [37,47] and provide competitive results. While other methods can be generalized to non-trivial unit cells [60] as well, in this work we focus on the CTMRG method due to its straightforward generalization and flexibility in handling arbitrary unit cells of size $(L_x, L_y)$. Furthermore, the CTMRG method provides a robust agnosticism towards possible non-Hermitian transfer operators, which has to be treated with more care in, e.g., boundary MPS methods.

The method we will present in this section gives rise to an upper bound of the ground state energy in the sense of the variational principle as stated in Eq. (2). But we wish to point out at this point that for that to be strictly true it would be necessary to choose the CTMRG refinement parameter $\chi_E$, introduced in detail in Sec. 2.2, to be $\chi_E \to \infty$. However, in practice we increase this refinement parameter $\chi_E$ until all observables are converged.

### 2.1 iPEPS setup

As introduced in the last section, we aim to simulate quantum many-body systems directly in the thermodynamic limit. To this end, we consider a unit cell of lattice sites that is repeated periodically over the infinite two-dimensional lattice. Reflecting this, the general configurations of the iPEPS Ansatz are defined with an arbitrary unit cell of size $(L_x, L_y)$ on the square lattice. The lattice setup, denoted by $\mathcal{L}$, can be specified by a single matrix, which uniquely determines the different lattice sites as well as their arrangement. Let us consider a concrete example of an $(L_x, L_y) = (2, 2)$ state with only two and all four individual tensors, denoted by

$$\mathcal{L}_1 = \begin{pmatrix} A & B \\ B & A \end{pmatrix}, \qquad \mathcal{L}_2 = \begin{pmatrix} A & C \\ B & D \end{pmatrix}. \tag{3}$$

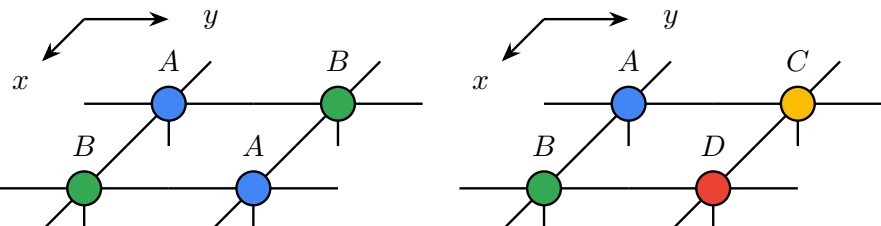

Figure 1: iPEPS ansätze with a unit cell of size $(L_x, L_y) = (2, 2)$ and only two (left) and four (right) different tensors as defined in Eq. (3).

The corresponding iPEPS ansätze are visualized in Fig. 1. Here, the rows/columns of $\mathcal{L}$ correspond to the $x/y$ lattice directions. The unit cell $\mathcal{L}$ is repeated periodically to generate the full two-dimensional system. As usual, the bulk bond dimension of the iPEPS tensors, denoted by $\chi_B$, controls the accuracy of the Ansatz. An iPEPS state with $N$ different tensors in the unit cell consists of $Np\chi_B^4$ variational parameters, which we aim to optimize such that the iPEPS wave function represents an approximation of the ground state of a specific Hamiltonian. The parameter $p$ denotes the dimension of the physical Hilbert space, e.g., $p = 2$ for a system of spin-1/2 particles.

The right choice of the unit cell is crucial in order to capture the structure of the targeted state. A mismatch of the Ansatz could not only lead to a bad estimate of the ground state, but also to no convergence in the CTMRG routine at all. Different lattice configurations have to be evaluated for specific problems to find the correct pattern.

To circumvent the problem of a fixed and a priori chosen unit cell structure, recently an alternative description to the periodic structure has been proposed [61]. This approach is applicable if the Hamiltonian has a certain global symmetry, where the additional degree of freedom can be employed to reduce the description of the state to a subspace, e.g. $SU(2)$ for spin-1/2 systems. Here the state is described by the smallest possible unit cell, i.e. a single site for a square lattice, as well as a product of local unitary operators parameterized by a wave vector $\mathbf{k} = (k_x, k_y)$. A fixed choice of the wave vector then corresponds to the specification of a unit cell structure in the common iPEPS setup. This approach allows for a variational optimization of the wave vector along with the translationally invariant iPEPS tensor, removing the need to choose a fixed unit cell structure altogether.

In this work we restrict the description of the method to the common iPEPS setup with not only trivial unit cells. This enables the adaption of the framework to arbitrary, in general non-symmetric Hamiltonian models.

## 2.2 CTMRG backbone

One major drawback of two-dimensional TNs such as iPEPS is that the contraction of the full lattice can only be computed approximately. This is due to complexity theoretic obstructions [19,20] and – practically speaking – the lack of a canonical form, which can only be found in loop-free tensor networks, for instance in matrix product states [8]. In order to circumvent the unfeasible exact contraction of the infinite 2d lattice, we employ an approximation scheme, the directional *corner transfer matrix renormalization group* (CTMRG) routine for iPEPS states with arbitrary unit cells of size $(L_x, L_y)$. The CTMRG method approximates the calculation of the norm $\langle \psi | \psi \rangle$ of the quantum state on the infinite square lattice by a set of effective environment tensors. This is achieved by an iterative coarse-graining procedure, in which all (local) iPEPS tensors in the unit cell $\mathcal{L}$ are successively absorbed into the environment tensors towards all lattice directions, until the environment converges to a fixed-point. We will present a summary of the directional CTMRG methods for an arbitrary unit cell, following

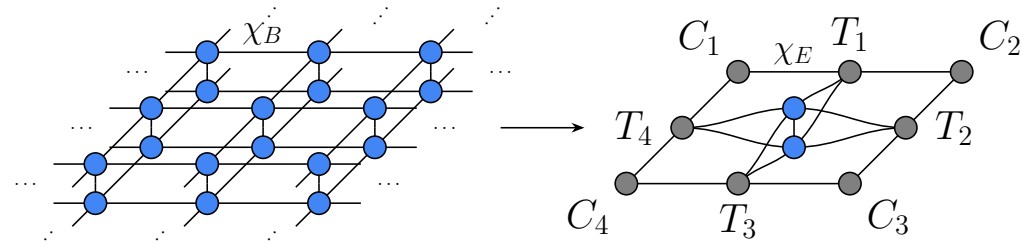

Figure 2: The norm of an iPEPS (here with a single-site unit cell) at a bulk bond dimension $\chi_B$ is approximated by a set of eight fixed-point environment tensors. The environment bond dimension $\chi_E$ controls the approximations in the CTMRG routine.

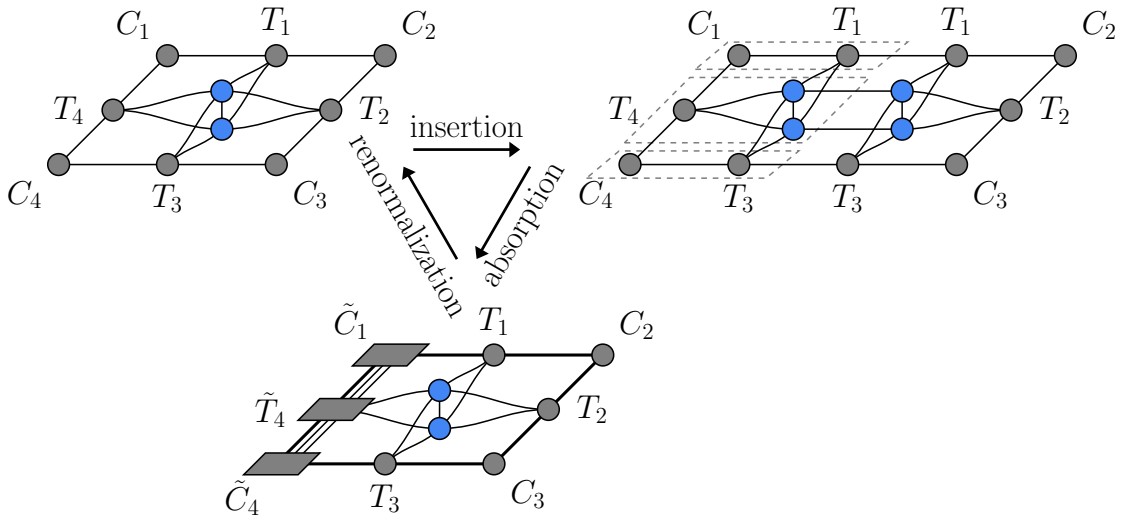

Figure 3: Main steps of a left CTMRG move. One column of tensors is inserted into the network. Upon absorption of these tensors, the environment bond dimension grows rapidly, requiring a renormalisation step.

the state-of-the-art procedure [62–64]. The effective environment is displayed in Fig. 2, here for simplicity for a square lattice with a single-site unit cell $\mathcal{L} = (A)$. It consists of a set of eight fixed-point tensors, four corner tensors $\{C_1, C_2, C_3, C_4\}$ as well as four transfer tensors $\{T_1, T_2, T_3, T_4\}$, the latter sometimes also called edge tensors. In case of a larger unit cell, such a set of eight environment tensors is computed for each individually specified iPEPS tensor in the unit cell. The unavoidable approximations in the environment calculations are controlled by a second refinement parameter, the environment bond dimension $\chi_E$.

In one full CTMRG step, the complete iPEPS unit cell is absorbed into the four lattice directions, such that the eight CTMRG tensors are updated for every iPEPS tensor. This is done column-by-column or row-by-row, depending on the direction. In each absorption step the environment bond dimension $\chi_E$ grows by a factor of $\chi_B^2$. To avoid an exponential increase in memory consumption and computation time, we need a method to truncate the bond dimension back to $\chi_E$. In order to do this, we calculate renormalization projectors for each row or column. Projectors are computed from a suitable patch of the iPEPS state including the effective environments, to find a best-possible truncation of the bond dimension. Different approaches for their calculations have been proposed in the literature, which we will discuss in detail below, especially in the context of AD. In the following description of the CTMRG procedure we focus on a left absorption move, which grows all left environment tensors $\{C_4, T_4, C_1\}$. The main steps of insertion, absorption and renormalization are shown in Fig. 3. In Sec. 2.2.1,

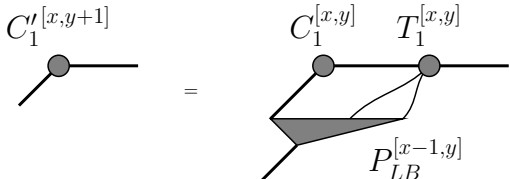

Figure 4: Update of the corner tensor $C_1$ in a left CTMRG step.

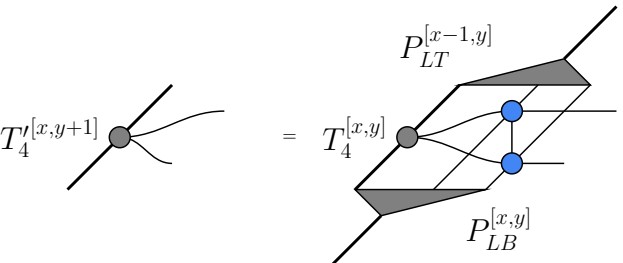

Figure 5: Update of the transfer matrix $T_4$ in a left CTMRG step. Here the projectors generally belong to different subspaces, unless the system is one-site translational invariant.

we will explain the full absorption procedure including renormalization, as it is done in practise. Although projectors need to be calculated before the absorption, their motivation and the calculation of different projects is discussed later in Sec. 2.2.2.

### 2.2.1 Absorption of iPEPS tensors

In order to generate the CTMRG environment tensors, such that they converge to a fixed-point eventually, the iPEPS tensors are absorbed into them. To this end, we start with the network of one iPEPS tensor in the unit cell and its accompanying environment tensors. This is depicted in Fig. 3 in the top left. As shown on the top right of this figure, the network is extended by inserting one column, consisting of an iPEPS tensor and the top and bottom transfer tensors. While we depict the case of a single-site unit cell in Fig. 3, we note that the column of tensors to be inserted is generally dictated by the unit cell structure of the iPEPS Ansatz, i.e., the left neighbor with the corresponding environment tensors for a left move. This crucial positional information for multi-site unit cells is specified by the coordinate superscripts in the descriptions below. As indicated by the dashed line in Fig. 3, we absorb the inserted column into the left environment tensors by contracting all left pointing edges. This yields new environment tensors whose bond dimensions have grown by a factor $\chi_B^2$ due to the virtual iPEPS indices, thus we need a way to truncate the dimension back to the CTMRG refinement parameter $\chi_E$. This is done using the projectors we will discuss and compute in the next section. For now we introduce them as abstract objects labeled $P$ that implement the dimensional reduction (i.e., the renormalization step) in an approximate but numerically feasible way. The updated tensor $C_1'$ is then given by the contraction in Fig. 4. As discussed before, the correct tensors and projectors have to be used in accordance with the periodicity of the unit cell. The iPEPS tensor is now absorbed into the left transfer matrix $T_4'$, where two projectors are needed to truncate the enlarged environment bond dimension. This is visualized in Fig. 5. Finally, the lower corner tensor $C_4'$ is updated, by absorbing a transfer matrix $T_3$ and using another projector. The three absorption steps in Figs. 4, 5 and 6 are performed for all rows $x$ at a fixed column $y$, before moving to the next column $y+1$. The process of computing projectors and growing the environment tensors is repeated for each column of the iPEPS unit

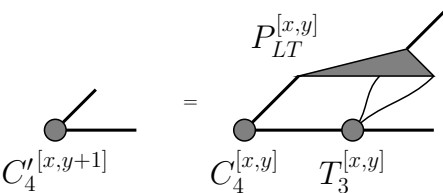

Figure 6: Update of the corner tensor $C_4$ in a left CTM step.

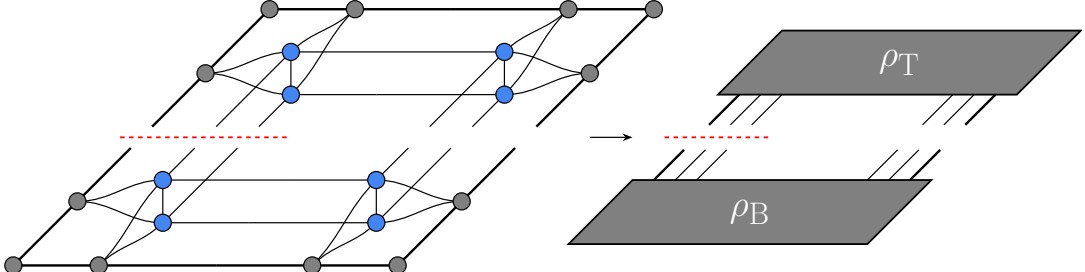

Figure 7: Network of $2 \times 2$ iPEPS tensors and the corresponding CTMRG tensors, used as a starting point to compute the truncation projectors. For a left CTMRG step the top and bottom part is contracted into the matrices $\rho_{\mathrm{T}}$ and $\rho_{\mathrm{B}}$ with dimension $(\chi_E \chi_B^2) \times (\chi_E \chi_B^2)$. The red dashed line indicates the bonds that are renormalized back to a bond dimension $\chi_E$.

cell, until the complete unit cell of $L_x \times L_y$ tensors has been absorbed into the left environment. This yields updated tensors $C_1'$, $T_4'$ and $C_4'$ for all $[x, y]$.

The absorption of a full unit cell is then performed for the other three directions. In a top move the tensors $C_1$, $T_1$ and $C_2$ are grown, in a right move the tensors $C_2$, $T_2$ and $C_3$ and in a bottom move the tensors $C_3$, $T_3$ and $C_4$. This completes a *single* CTMRG step, which is then repeated in the directional procedure until convergence is reached. In Sec. 2.2.3 we discuss appropriate convergence measures.

### 2.2.2   Calculation of projectors

In order to avoid an exponential increase of the bond dimension while growing the environment tensors, projectors are introduced to keep the bond dimension at a maximal value of $\chi_E$. Here, we will describe a common scheme to compute those projectors [63] and discuss some properties of their use in combination with AD [42]. The task of finding good projectors essentially comes down to finding a basis for the virtual space, whose bond dimension we aim to reduce, that can be used to distinguish between "more and less important" sub-spaces. This way, we can ideally reduce the dimension while keeping the most important sub-space. In what follows, we consider the lattice environment of the virtual space that we aim to truncate using the CTMRG environment tensors. To this end, we use a *singular value decomposition* (SVD) to identify the basis, in which the bond is optimally truncated such that we keep the most relevant information of this lattice environment. The lattice environment that we consider is shown in Fig. 7, where the red dotted line identifies the bonds that we aim to optimally truncate, illustrated for the example of a left absorption step. The arrangement of the tensors in the network of Fig. 7 follows the unit cell definition $\mathcal{L}$. For the trivial, single-site unit cell $\mathcal{L} = (A)$, all four iPEPS tensors are the same. We note that for a larger unit cell, cf. Fig. 1, the iPEPS tensors and their adjacent environments have to be chosen according to its periodicity. This setup for the arrangement is favorable, since it incorporates the (approximated) effect of the infinite environment by including all CTM tensors for the different lattice directions.

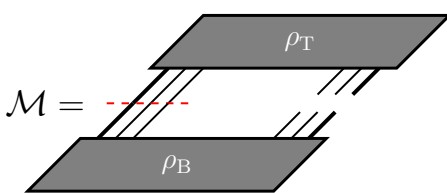

Figure 8: Matrix $\mathcal{M}$ as defined by Eq. (4) in graphical TN notation. The red dashed line indicates the bonds that are renormalized back to a bond dimension $\chi_E$.

The projectors are used to renormalize the three left open tensor indices with combined bond dimension $\chi_E \chi_B^2$ back to the environment bond dimension $\chi_E$ in a left absorption step. In order to compute them, we start by defining the matrix

$$\mathcal{M} = \rho_{\mathrm{B}} \cdot \rho_{\mathrm{T}}, \tag{4}$$

that represents the lattice environment of the virtual bond that we would like to truncate, as visualized in Fig. 8.

The procedure outlined here aims to find projectors $P_{LT}$ and $P_{LB}$, such that the truncated matrix,

$$\mathcal{M}_{\mathrm{trunc}} = \rho_{\mathrm{B}} \cdot P_{LT} \cdot P_{LB} \cdot \rho_{\mathrm{T}}, \tag{5}$$

is an optimal approximation to $\mathcal{M}$. To achieve this, we perform a singular value decomposition on $\mathcal{M}$, i.e.,

$$\mathcal{M} = U_L S_L V_L^\dagger. \tag{6}$$

This factorization introduces a basis which allows for a separation of more relevant and less relevant sub-spaces. To this end, we choose the largest $\chi_E$ singular values and their corresponding singular vectors for the construction of the projectors. Furthermore, we define

$$S_L^+ = \mathrm{inv}\left(\sqrt{S_L}\right), \tag{7}$$

where a pseudo-inverse with a certain tolerance is used. To increase the numerical stability, a threshold of typically $10^{-6}$ (corresponding to a threshold of $10^{-12}$ for the singular values) is used. Smaller singular values are set to zero. The use of a pseudo-inverse in the generation of the projectors is equivalent to the construction of a projector with lower environment bond dimension. Finally, the projectors to renormalize the left absorption step are construced as

$$\begin{aligned} P_{LT} &= \rho_T \cdot V_L \cdot S_L^+, \\ P_{LB} &= S_L^+ \cdot U_L^\dagger \cdot \rho_B. \end{aligned} \tag{8}$$

Here $\rho_T$ and $\rho_B$ again denote the top and bottom part of $\mathcal{M}$ as introduced in Fig. 7. We would like to point out the fact that without a truncation in the SVD above, the product of the projectors we create in this way assembles the identity

$$\begin{aligned} P_{LT} \cdot P_{LB} &= \rho_T \cdot V_L \cdot S_L^{-1} \cdot U_L^\dagger \cdot \rho_B \\ &= \rho_T \cdot (\rho_{\mathrm{B}} \cdot \rho_{\mathrm{T}})^{-1} \cdot \rho_B = \mathbb{1}. \end{aligned} \tag{9}$$

We stress again, that the choice of truncation in the calculations of the projectors is optimal in order to approximate the lattice environment $\mathcal{M}$. A graphical representation of these projectors is given in Fig. 9.

During a left-move, described in the previous section, we absorb the iPEPS tensors in the unit cell column-by-column into the left environments. A renormalization step is required for

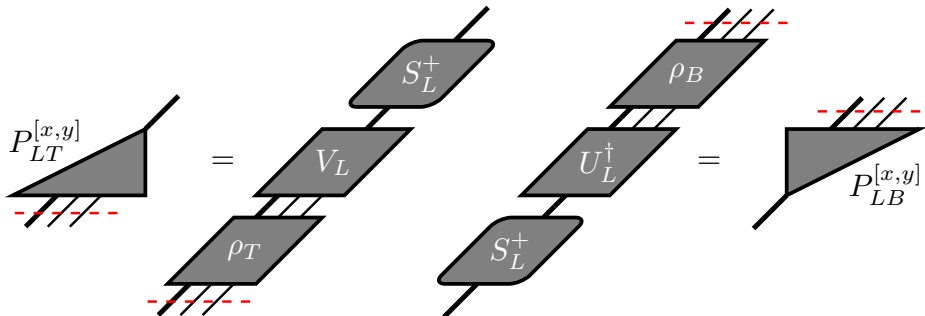

Figure 9: Calculation of top and bottom projectors for a left CTMRG absorption step. The red dashed line indicates the bonds that are renormalized back to a bond dimension $\chi_E$.

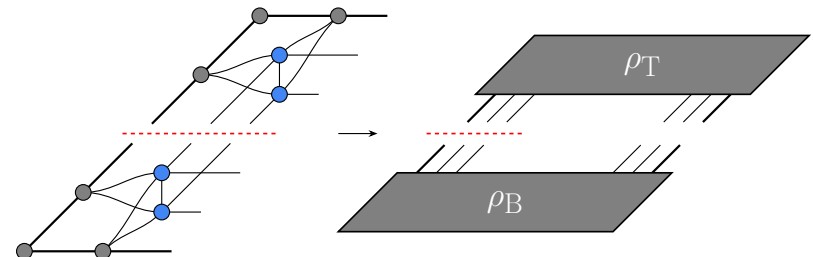

Figure 10: Network of $2 \times 1$ iPEPS tensor and the corresponding CTMRG tensors, which is used as a reduced network to calculate the half projectors for a left CTMRG step. The red dashed line indicates the bonds that are renormalized back to a bond dimension $\chi_E$.

each of those moves, resulting in projectors that are specific to every bond. We therefore label them by the positions in the unit cell, i.e., $P_{LT}^{[x,y]}$ and $P_{LB}^{[x,y]}$.

The process to generate the projectors described above uses the full lattice environment $\mathcal{M}$, and thus we call them *full projectors*. It should be noted that Fishman et al. have proposed a scheme to calculate equivalent projectors in a fashion that is numerically more stable, at the cost of being computationally more expensive [64]. Their method is particularly useful in the case of a singular value spectrum of $\mathcal{M}$ that decays very fast.

Finally, different lattice environments of the virtual bond in question can be used to generate projectors. A very practical version are the so called *half projectors*. For those we choose a lattice environment as illustrated in Fig. 10. These projectors are computationally less costly, as they require a smaller network to be contracted. They only take into account correlation within one half of the network, however this proves to be sufficient in many different applications. Lately, there have been proposals for even cheaper alternatives of lattice environments and projector calculations [65], which yet have to be tested in the context of automatic differentiation and variational iPEPS optimization.

### 2.2.3 Convergence and CTMRG fixed-points

The CTMRG routine as described above is a power-method that eventually converges to a fixed-point. At this fixed-point, the set of environment tensors describes the contraction of the infinite lattice with an approximation controlled by the environment bond dimension $\chi_E$. Convergence of the CTMRG tensors to the fixed-point can be monitored in different ways. In regular applications (those that do not involve automatic differentiation and gradients) the singular value spectrum of the corner tensors is typically a good quantity. Once the norm differ-

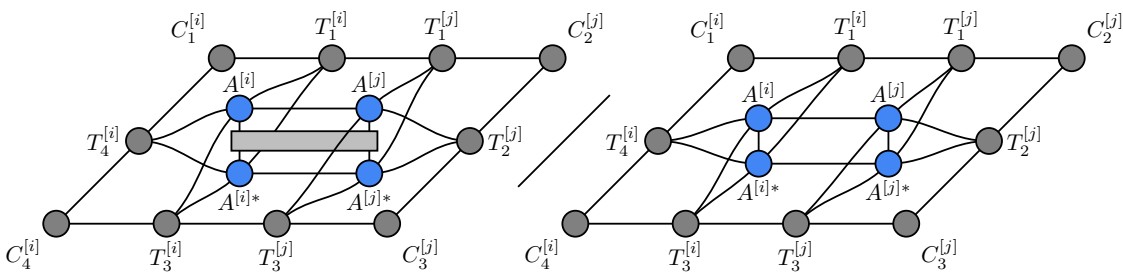

Figure 11: Expectation values of a (horizontal) nearest-neighbour Hamiltonian term $\langle\psi|h_{i,j}|\psi\rangle / \langle\psi|\psi\rangle$ in tensor network notation, using the fixed-point CTMRG environments.

ence of the spectrum between two successive CTM steps converges below a certain threshold, the environment tensors are assumed to be converged.

One peculiarity that is however not incorporated in this convergence check is sign or phase fluctuation for real or complex tensor entries, respectively. This means that, while projectors and hence the CTMRG tensors converge in absolute value, their entries can have different signs/phases in consecutive CTM steps. For reasons that become clear in Sec. 2.5 it is however required to reach *element-wise convergence* in the environment tensors for them to represent an actual fixed-point [42]. Those fluctuations originate from the gauge freedom in the SVD performed in Eq. (6). This is reflected in the freedom of introducing a unitary (block-)diagonal matrix $\Gamma$ in an SVD,

$$\mathcal{M} = USV^{\dagger} = (U\Gamma)S\left(\Gamma^{\dagger}V^{\dagger}\right), \tag{10}$$

which leaves the expression invariant. The gauge freedom from the SVD directly affects the calculation of the projectors, such that we aim to fix the phases while computing these projectors. By eliminating this gauge freedom, at the true fixed-point, both projectors and environment tensors should be converged element-wise.

To fix the gauge, we introduce a diagonal unitary matrix $\Gamma$ that redefines the phase of the largest entry (in absolute value) of every left singular vector to place it on the positive real axis [42]. To avoid instabilities of this gauge-fixing procedure due to numerical quasi-degeneracies, we always pick the first of such largest elements in basis order. Other choices, like addressing the first element with magnitude above a fixed threshold, are also possible. We further note that an alternative scheme to archive a fixed point in the CTMRG has recently been proposed [66].

## 2.3 Energy expectation values

Computing the energy expectation value required for the energy minimization is straightforward using the CTMRG environment tensors. Assuming a Hamiltonian with only nearest-neighbour interaction terms, individual bond energies can be computed as shown in Fig. 11. The full energy expectation value, $\langle\psi|H|\psi\rangle / \langle\psi|\psi\rangle$, is obtained by collecting all different energy contributions, i.e., all different terms in the Hamiltonian. Longer-range interaction can be treated as well, by simply enlarging the diagrams of Fig. 11 and performing more expensive contractions, which however occur only once per optimization step. In order to formulate a variational optimization of the tensor coefficients parametrizing the wave function, a gradient for the energy expectation value – including the foregone fixed-point CTMRG routine – is required. This is achieved by the concept of automatic differentiation, as we will describe next.

Figure 12: Example of a computational graph for the function decomposition in Eq. (11).

## 2.4 Automatic differentiation

*Automatic differentiation* (AD), sometimes also referred to as *algorithmic differentiation* or *automated differentiation*, is a method for taking the derivative of a complicated function which is evaluated by some computer algorithm. It has been an important tool for optimization tasks in machine learning for many years. An introduction can be found in e.g. Ref. [67]. After its initial introduction in a foundational work [37], AD has found increasing applications in numerical TN algorithms in recent years [38, 39, 41, 42, 44, 45]. For the sake of simplicity, let us consider a function $E : \mathbb{R}^n \to \mathbb{R}^m$ for which we would like to evaluate the derivative. Noticeably, extensions to complex numbers are possible, and we provide some additional comments in Appendix B. We have the particular use-case of the energy expectation value $E(|\psi\rangle) = \langle\psi|H|\psi\rangle / \langle\psi|\psi\rangle$ of an iPEPS in mind, in which case the co-domain of the function $E$ is $\mathbb{R}$. As we explain below, this has some important consequences for the use of AD.

Automatic differentiation makes use of the fact that many functions and algorithms are fundamentally built by concatenating elementary operations and functions like addition, multiplication, projection, exponentiation and taking powers, whose derivatives are known. The central insight is now that we can build up the gradient of a more complicated function from the derivatives of its elementary constituents by the *chain rule of differentiation*. In principle this even allows for a computation of the gradient to machine precision. It should be noted however, that it is neither necessary nor useful to deconstruct every function into its most elementary parts. Rather it is advantageous to deconstruct the function at hand only into a minimal amount of constituent-functions for which a derivative can be determined. These functions are often referred to as the *primitives* of the function of interest $E$. Primitives might themselves be a composition of many constituents but the derivative of the primitives themselves is known as a whole. An illustrative example for a primitive is a function that takes two matrices as an input and outputs the multiplication of them. On an elementary level this function is composed out of many multiplications and additions, but one can write down the derivative w.r.t. its inputs immediately. The choice of primitives describes the level of coarseness on which the AD process needs to know the details of the function $E$ to compute the desired gradient. Defining large primitives of a function can reduce memory consumption, as well as increase performance and numerical stability of the AD process, e.g., by avoiding spurious divergences. Once the high-level function $E$ has been decomposed into its minimal number of primitives, we can represent this decomposition with a so called *computational graph*. The computational graph is a directed, a-cyclic graph whose vertices represent the data generated as intermediate results by the primitives and the edges represent the primitives themselves, that transform the data from input to output.

As an example let us suppose we are able to decompose the function $E$ into three primitives $f_1$, $f_2$ and $f_3$, such that $E = f_3 \circ f_2 \circ f_1$. The primitives are maps between intermediate spaces

$$E : \mathbb{R}^{n_1} \xrightarrow{f_1} \mathbb{R}^{n_2} \xrightarrow{f_2} \mathbb{R}^{n_3} \xrightarrow{f_3} \mathbb{R}^{n_4} , \tag{11}$$

and we refer to the variables in theses spaces as $\vec{x}_i \in \mathbb{R}^{n_i}$. The computation graph illustrating this situation is shown in Fig. 12. AD can be performed in two distinct schemes, often called

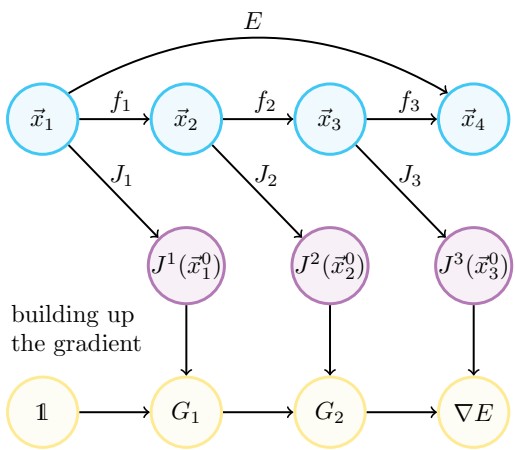

Figure 13: Illustration of forward-mode AD as described in Eq. (14) for the function decomposition in Eq. (11).

*forward-* and *backward-mode* AD. In the following we will demonstrate the two AD modes with the example of our previously introduced function $E$ and its primitives. This will also serve to illustrate the computational cost of these AD schemes for the iPEPS use-case. Since $f_1, f_2$ and $f_3$ are said to be primitives, their Jacobians

$$
J^i \; : \; \mathbb{R}^{n_i} \longrightarrow \mathbb{R}^{n_{i+1}} \times \mathbb{R}^{n_i},
$$
$$
J^i(\vec{x}_i^0) = \left(\frac{\partial f_i}{\partial \vec{x}_i}\right)\bigg|_{\vec{x}_i = \vec{x}_i^0}, \tag{12}
$$

are known. An AD evaluation of the gradient of $E$ at a specific point $\vec{x}_1^0$ is then given by the chain rule, the concatenation of the Jacobians of the primitives

$$
\nabla E(\vec{x}_1^0) = J^3(\vec{x}_3^0) \cdot J^2(\vec{x}_2^0) \cdot J^1(\vec{x}_1^0), \tag{13}
$$

with $f_i(\vec{x}_i^0) = \vec{x}_{i+1}^0$. The difference between the *forward-* and *backward-mode AD* essentially comes down to the question from which side we perform the multiplication of the Jacobians above.

In the *forward-mode* AD scheme, the gradient is built up simultaneously with the evaluation of the primitives $f_1, f_2$ and $f_3$, according to the prescription

$$
f_i(\vec{x}_i^0) = \vec{x}_{i+1}^0,
$$
$$
G_i = J^i(\vec{x}_i^0) \cdot G_{i-1}, \tag{14}
$$

for the $i$-th step, with the starting condition $G_0 := \mathbb{1}_{n_1 \times n_1}$ and with the final result that is given by $G_3 = \nabla E(\vec{x}_1^0) \in \mathbb{R}^{n_4 \times n_1}$. We see that in this case we build up Eq. (13) from right to left or "along the computational graph" as illustrated in Fig. 13. At first sight, such a procedure offers the potential advantage of not requiring to store intermediate results of the primitives in memory. However, if the dimension of the input (domain of $E$) is much larger than the dimension of the output (co-domain of $E$) – as it is the case in our use-case of iPEPS – this procedure becomes computationally very heavy. Indeed, saving and multiplying the large Jacobians in Eqs. (14) is often impractical. Thus, it is common to split up the starting condition $G_0 := \mathbb{1}_{n_1 \times n_1}$ into the $n_1$ canonical basis vectors $\{\vec{e}_i\}_{i=1,\dots,n_1}$. The procedure to generate the gradient from Eq. (14) is then repeated $n_1$ times, each iteration generating a single component $i$. In this case, each step of the process of generating a component of the gradient is done by calculating a *Jacobian-vector product* (JVP), so that only the resulting vector has to be stored.

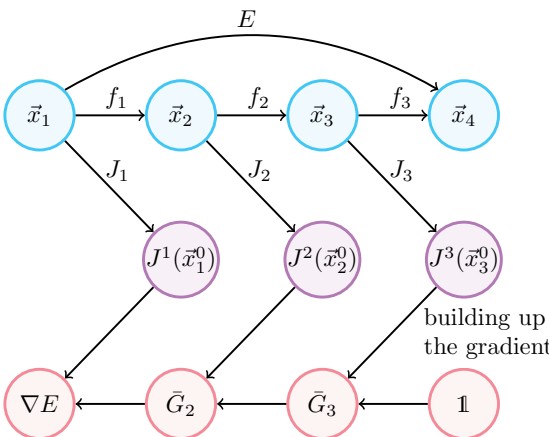

Figure 14: Illustration of backward-mode AD as described in Eq. (15) for the function decomposition in Eq. (11).

In order to create the full gradient in this way we need to repeat the procedure $n_1$ times, and the cost of calculating the full gradient scales as $\mathcal{O}(n_1) \times \mathcal{O}(E)$, where $\mathcal{O}(E)$ is the cost of evaluating $E$.

The *backward-mode* AD scheme works instead by first evaluating the function $E$ and storing all intermediate results of the primitives along the way, and by then applying the iterative prescription

$$\bar{G}_i = \bar{G}_{i+1} \cdot J^i(\vec{x}_i^0), \tag{15}$$

with the starting condition $\bar{G}_4 = \mathbb{1}_{n_4 \times n_4}$ and the final result $\bar{G}_1 = \nabla E(\vec{x}_1^0) \in \mathbb{R}^{n_4 \times n_1}$. In the AD literature the objects $\bar{G}_i$ are called adjoint variables and the functions that map the adjoint variable on to each other, defined by Eq. (15), are called adjoint functions. We refer to Appendix A for more details on the adjoint functions and adjoint variables. In some parts of the literature the adjoint functions are also called pullbacks, which can be understood by looking at AD in language of differential geometry, cf. Appendix D. We see that in this case we build up Eq. (13) from left to right or as graphically illustrated in Fig. 14. This scheme has the advantage of being computationally much cheaper if the output (co-domain) dimension is smaller than the input (domain) dimension – precisely the situation of our iPEPS setup, with $n_1 = N p \chi_B^4$ and $n_4 = 1$. We indeed only need to compute *vector-Jacobian products* (VJP) when evaluating the gradient, and, moreover, the full gradient is computed at once, instead of just a single element at a time as in the forward-mode AD scheme. This is why the cost of calculating the gradient of the energy expectation value with *backwards-mode* AD is $\mathcal{O}(1) \times \mathcal{O}(E)$, which is superior to the cost of *forward-mode* AD. However, since we need to save all intermediate results of the primitives along the way in order to compute the gradient, the memory requirement for this scheme is in principle unbounded. Fortunately, the fixed-point condition for the iPEPS environments can be used to guarantee that the memory remains bounded in our calculations, as we illustrate in the following section.

## 2.5 Calculation of the gradient at the CTMRG fixed-point

Computationally, the CTMRG routine represents the bottleneck of the full iPEPS energy function. It involves many expensive contractions and SVDs. Moreover, it requires an a priori unknown number of CTMRG iterations to reach convergence of the environment tensors. This would be especially disadvantageous for the gradient evaluation using plain-vanilla backward-mode AD, since this would require unrolling all the performed CTMRG iterations and paying a memory consumption linear in their number. However, this can be avoided by leveraging

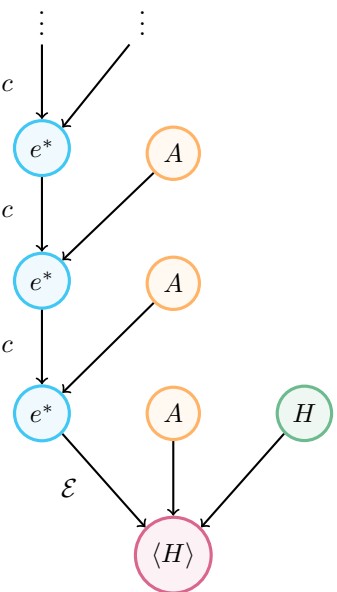

Figure 15: Computational graph of the CTMRG procedure for calculating the energy density at fixed point.

that fact that the CTMRG iteration eventually converges to a fixed point, and this is precisely the condition under which the energy evaluation is then performed. As soon this fixed point is reached, all CTMRG iterations are identical, i.e., reproducing the converged environment tensors. We can, in this situation, get away with only saving intermediate results from such a converged CTMRG iteration. This reduces the memory requirements by a factor of the number of CTMRG iterations that we perform [37]. We stress here that, for this approach to work, we must make sure that the CTMRG procedure reaches an actual fixed point, meaning that all CTMRG environment tensors are converged element wise as discussed in Sec. 2.2.3. The fixed-point equation can be written as

$$e^*(A) = c(A, e^*(A)),\qquad(16)$$

where the function $c$ is one full CTMRG iteration, $A$ are the iPEPS tensors which are constant during the CTMRG procedure and $e^*(A)$ represents the CTMRG environment tensors at the fixed-point. $\mathcal{E}$ is the function that maps the iPEPS tensors with the fixed point environment tensors and the Hamiltonian operators to the energy expectation value. The computational graph for the ground state energy is illustrated in Fig. 15. From it we can construct the form of the gradient of the energy expectation value with respect to the parameters of the iPEPS tensors $A$,

$$\frac{\partial \langle H \rangle}{\partial A} = \frac{\partial \mathcal{E}}{\partial A} + \frac{\partial \mathcal{E}}{\partial e^*} \sum_{n=0}^{\infty} \left( \frac{\partial c}{\partial e^*} \right)^n \frac{\partial c}{\partial A}.\qquad(17)$$

In practice this infinite sum is evaluated to finite order until the resulting gradient is converged to finite accuracy. An alternative viewpoint on the gradient at the fixed-point of the CTMRG procedure is presented in the Appendix C.

It has recently been noted in Ref. [66] that the stability and accuracy of the SVD derivative can be improved by including a previously neglected gradient contribution from the truncated part of the singular value spectrum.

## 2.6  Optimization

As discussed in the introduction of Sec. 2 we seek to find the iPEPS approximation $|\psi\rangle_{\text{TN}}$ of the ground state vector $|\psi_0\rangle$. Employing the methods discussed in the last sections we can describe this energy calculation as function $E(|\psi\rangle_{\text{TN}})$, consisting of the CTMRG power-method and the expectation value approximation using the resulting CTMRG environment tensors. Since we can calculate the gradient $\nabla E(|\psi\rangle_{\text{TN}})$ of this real scalar function it is straightforward to use well-known optimization methods to find the energy minimum. We would like to stress that the state vector $|\psi\rangle_{\text{TN}}$, and thus the energy function, only depends on the tensors defining the iPEPS Ansatz and not the environment tensors since they are implicitly calculated from the Ansatz. In this discussion we focus on two types of methods based on the gradient: The *(nonlinear) conjugate gradient* (CG) [68–72] and the quasi-Newton methods [73–78].

A naive approach to find the minimum of a function $E(|\psi_i\rangle)$, of which the gradient $\nabla E(|\psi_i\rangle)$ is known, is to shift the input parameters $|\psi_i\rangle$ sufficiently along the negative gradient so that we find a new position $|\psi_{i+1}\rangle$ where the function value is reduced. At the end of this section we discuss what a sufficient step size means in this context. Iterating this procedure to a point where the gradient of the function vanishes (within a pre-defined tolerance) yields a solution to the optimisation problem. Thus either a saddle point or a (local) minimum is reached then. This method is called steepest gradient descent. Although it resembles one of the simplest methods to find a descent direction, it is known to have a very slow convergence for difficult problems, e.g., for functions with narrow valleys [79]. Therefore, we use in practice more sophisticated methods to determine the descent direction.

The family of nonlinear conjugate gradient as generalization of the linear conjugate gradient method modifies this approach. Instead of using the negative gradient as a direction in each iteration step it uses a descent direction which is conjugated to the previous ones. For the linear conjugate gradient method there is a known factor $\beta_i$ to calculate the new descent direction $d_i = g_i + \beta_i d_{i-1}$ from the gradient $g_i$ of the current step and the descent direction $d_{i-1}$ of last step. In the generalization for nonlinear functions this parameter is not uniquely determined anymore, however there are different approaches to estimate this parameter in the literature [69–71]. In our implementation we chose the nonlinear conjugate gradient method in the formulation as has been suggested by Hager and Zhang [72],

$$\begin{aligned}
\tilde{\beta}_i^{\text{HZ}} &= \frac{1}{d_{i-1}^{\mathsf{T}} y_i} \left( y_i - 2 d_{i-1} \frac{\|y_i\|^2}{d_{i-1}^{\mathsf{T}} y_i} \right)^{\mathsf{T}} g_i, \\
\eta_i &= \frac{-1}{\|d_{i-1}\| \min(\eta, \|g_{i-1}\|)}, \\
\beta_i^{\text{HZ}} &= \max(\tilde{\beta}_i^{\text{HZ}}, \eta_i),
\end{aligned} \tag{18}$$

with $\|\cdot\|$ the Euclidian norm, $y_i = g_i - g_{i-1}$ and $\eta > 0$ a numerical control parameter which has been set to $\eta = 0.01$ in the work by Hager and Zhang. In our tests and benchmarks this choice for $\beta_i$ has been proven to be numerically stable.

The other family of optimization methods we use in our implementation are the quasi-Newton methods, concretely the *Broyden–Fletcher–Goldfarb–Shanno* (BFGS) algorithm [75–78] and its *low-memory* (L-BFGS) variant [74, 80]. These methods are based on the Newton method where the descent direction is calculated using not only the gradient, but also the second derivative (the Hessian matrix). Unfortunately, it is computationally expensive to calculate the Hessian for large sets of input parameter, which makes this method only feasible for small parameter sets (i.e., iPEPS ansätze with a small number of variational parameters). Quasi-Newton methods solve this problem by not calculating the full Hessian, but an approximation of it. To this end, the gradient information from successive iteration steps is used to

update the approximation in each step. The BFGS algorithm stores the full approximated Hessian matrix, including the information from all previous steps. In contrast, the L-BFGS method calculates the effective descent direction in an iterative manner from the last $N$ optimization steps. This way not the full (approximated) Hessian has to be stored in memory but only the gradients of the last $N$ steps. This reduces the memory consumption by an order of magnitude. The disadvantage is that not the full information of all previous steps is considered, but only a fraction of it. Nevertheless, due to the memory requirements to store the full approximated Hessian in the standard BFGS method for larger iPEPS bond dimensions we use L-BFGS as the default quasi-Newton method.

As noted before, we would like to shift the variational parameters $x_i$ along the descent direction $d_i$ determined by the different algorithms discussed above. With this shift we aim to find a new Ansatz $x_{i+1} = x_i + \alpha_i d_i$ with $\alpha_i$ the step size along the descent direction. Ideally, we would like to find the optimal step size $\alpha_i = \min_\alpha E(x_i + \alpha d_i)$ minimizing the function value along the descent direction. However, determining this optimal value is computationally expensive and thus in practice, we stick to a sufficient step size fulfilling some conditions. The procedure to find this step size is called *line search* [81–84]. In our implementation we use the Wolfe conditions [82–84], since they guarantee properties which are feasible particularly for the (L-)BFGS method and its iterative update of the effect of the approximate Hessian.

## 2.7   Handling of physical symmetries

One important property of tensor networks is their ability to incorporate physical symmetries, such as global internal symmetries or spatial symmetries, into their structure exactly. This can be achieved at the level of each individual tensor, ensuring that the entire many-body wave function remains symmetry-invariant. On one hand, this approach allows for the targeting of specific sectors of the Hamiltonian and facilitates a symmetry-resolved analysis of the model under consideration. On the other hand, it restricts the number of remaining variational parameters and can lead to a substantial computational speed-up, which, in turn, enables access to larger bond dimensions.

Global symmetries can be implemented by making the tensors quantum number-preserving, with quantum numbers corresponding to the underlying symmetry group. This results in a sparse tensor block structure, where linear algebra operations are performed on a generally larger set of much smaller individual tensors, leading to more efficient manipulations. Common symmetries exploited in tensor network simulations include Abelian symmetries such as $\mathbb{Z}_N$ or $U(1)$ [85, 86], non-Abelian symmetries such as $SU(2)$ [87, 88], and also fermionic symmetries [89, 90]. Moreover, lattice symmetries can be directly implemented as well, e.g. by imposing reflection or rotation symmetry for the tensor entries.

By making the full set of algorithmic tensor operations, such as tensor initialization, contraction, factorization, index permutation, etc., aware of the block structure, symmetric tensors can be treated analogously to non-symmetric ones on a conceptual level. This includes their use in variational optimization based on automatic differentiation. There exist several open source libraries that can handle internal (non-)Abelian and lattice symmetries in TNs [91–94], some already within the scope of variational PEPS optimization [50].

## 2.8   Pitfalls and practical hints

### 2.8.1   Iterative SVD algorithm

We also advertise the use of iterative algorithms for the calculation of the SVD in the CTMRG procedure. This can be quite advantageous computationally, since only $\chi_E$ singular values are needed for a matrix of size $(\chi_E \chi_B^2) \times (\chi_E \chi_B^2)$ during the CTMRG. To this end, we use the

*Golub-Kahan-Lanczos* (GKL) bidiagonalization algorithm with additional orthogonalization for the Krylov vectors. This algorithm is available, e.g., in packages like KRYLOVKIT.JL [95] or ITERATIVESOLVERS.JL [96] in the JULIA programming language. We highlight the utility of this type of algorithm for the calculation of the SVD with the comparison of the computational time of the different algorithms in the iPEPS use case in Fig. 16.

### 2.8.2 Stability of the CTMRG routine

One of the basic prerequisite for a stable variational iPEPS optimization is a robust CTMRG routine fulfilling the convergence requirements discussed in Sec. 2.2.3. Obviously, there is the environment bond dimension $\chi_E$ to control the accuracy of the approximation of the environment. If the environment bond dimension is chosen too low, the approximation is invalid and the CTMRG routine can yield an inaccurate result for the expectation value. This could further lead to an unstable variational update. To check heuristically whether the refinement parameter $\chi_E$ is chosen sufficiently high, one can check the singular value spectrum obtained during the projector calculation as described in Sec. 2.2.2. As a reliable criteria for the amount of information loss, we compute the truncation error $\varepsilon_T$ given by the norm of the discarded singular values of the normalized spectrum [97]. If the truncation error is larger than some threshold (e.g., $\varepsilon_T > 10^{-5}$), one can assume that the environment bond dimension is chosen too low and has to be increased. Employing this procedure, the bond dimension can automatically be increased during the variational optimization if necessary. A sufficiently large $\chi_E$ is crucial as the AD optimization can otherwise exploit the inaccuracies of the CTMRG procedure, leading to false ground states with artificially low energy.

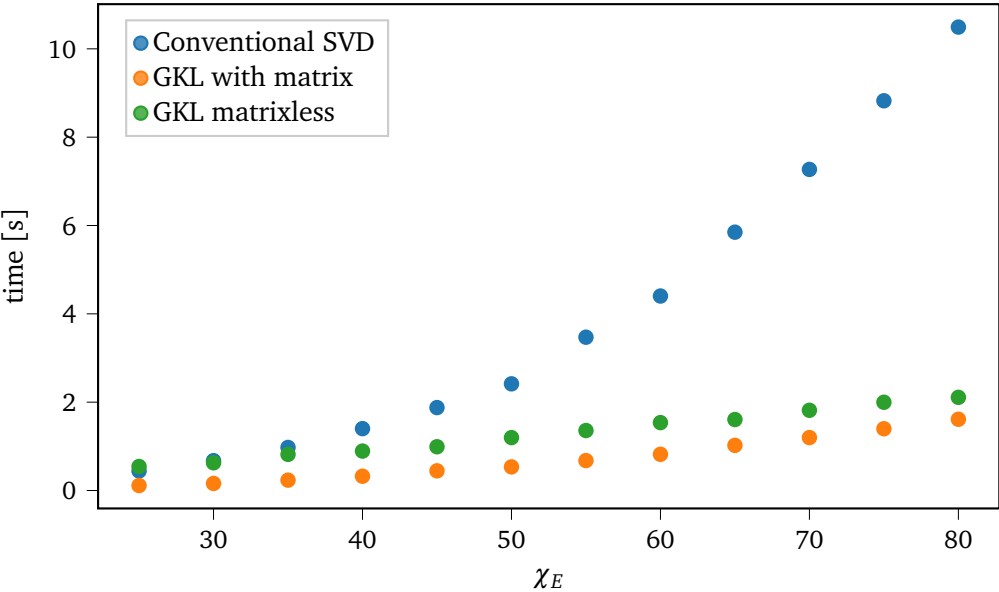

Figure 16: Comparison of the computational time for the calculation of the first $\chi_E$ singular values/vectors of a matrix of dimension $(\chi_E \chi_B^2) \times (\chi_E \chi_B^2)$ obtained in a CTMRG procedure with bond dimension $\chi_B = 6$. The conventional SVD (blue), which is truncated only after calculating the full SVD spectrum is substantially slower than the iterative GKL methods. The GKL algorithm in the CTMRG use case was showed comparable performance when constructing the $\chi_E \chi_B^2$ matrix explicitly (orange) or by just implementing its action of a vector (green). While the GKL algorithm for the case at moderate $d$ and $\chi_E$ constructing the matrix usually is faster, at larger $\chi_B$ and $\chi_E$ it can become advantageous to only implement the action of the matrix.

### 2.8.3 Prevention of local minima

An ideal iPEPS optimization finds the global energy minimum of the input Hamiltonian within the iPEPS Ansatz class of fixed unit cell and bond dimension. In practice, however, it is possible – and likely – that the algorithm gets stuck in local minima. In order to avoid local minima and reach the global optimum, there are a number of possible tricks. The naive way is to start several simulations with different random initial states. This is typically a practicable solution, although it is not well controllable and requires large computational resources.

An optimization of a system with a tendency for local minima might still be successful, if a suitable initial state is provided. One possibility are initial states obtained by imaginary-time evolution methods (simple update, full update [22,23,98]). While this is typically a convenient solution, it is sometimes necessary to perturb the input tensors with a small amount of noise (e.g., $10^{-2}$ in relative amplitude) to actually avoid local minima. As an alternative, one can input a converged state obtained from energy minimization of a different TN Ansatz, provided there is a suitable mapping between the different structures. Examples for this technique are provided for benchmarks on different lattices in Sec. 4.

Finally, the method of perturbing a suitable initial state with small amount of random noise of course could also be applied to the result of one optimization run. As suggested in the literature [99], this could help to escape possible local minima. Therefore, one could retry this method a few times and keep the best result of all runs.

### 2.8.4 Recycling of environments

The calculation of the environment tensors with the CTMRG routine is expensive and time consuming. During an optimization process one can reuse the environment tensors of the previous optimization step as input for the next. This is advisable in the advanced stages of the optimization, in which the gradient is already small. In this scenario the iPEPS tensors usually only change minutely, such that starting the CTMRG routine from the environments of the last PEPS tensor can reduce the number of CTMRG steps required for convergence substantially.

### 2.8.5 Analysing iPEPS data at finite bond dimensions

Data generated with the variational iPEPS setup inevitably carries finite iPEPS bond dimension $\chi_B$ (or even finite environment bond dimension $\chi_E$) effects. Several schemes are available to utilize the correlation length of the optimal tensors at a certain value of $\chi_B$ to extrapolate the values of observables [100–102]. Additionally, a extrapolation scheme using data of an optimized iPEPS state at finite $\chi_B$ and finite but suboptimal $\chi_E$ has been proposed and shown useful [103].

### 2.8.6 Degenerate singular values

Although very rare, a degenerate singular value spectrum in the calculation of the projectors can be an obstacle. The gradient of the SVD becomes ill-defined in this case, due to terms $F_{i,j} = 1/(s_j^2 - s_i^2)$ in the derivative [45], where $s_i$ are the singular values. Naturally, it would be desirable to remove the degeneracy by constraining the system to the correct physical symmetry, thereby grouping the degenerate singular values to common multiplets of the underlying symmetry group. If this is not possible or the degeneracies appear independently of a symmetry ("accidental" degeneracy), workarounds have to be used. One possibility is to add a small amount of noise in the form of a diagonal matrix $XX^{-1}$ on the CTMRG environment links, with the elements of $X$ drawn from a tiny interval $[1 - \varepsilon, 1 + \varepsilon]$. This can space out the singular value spectrum and stabilize the SVD derivative [104]. Recently an alternative procedure to

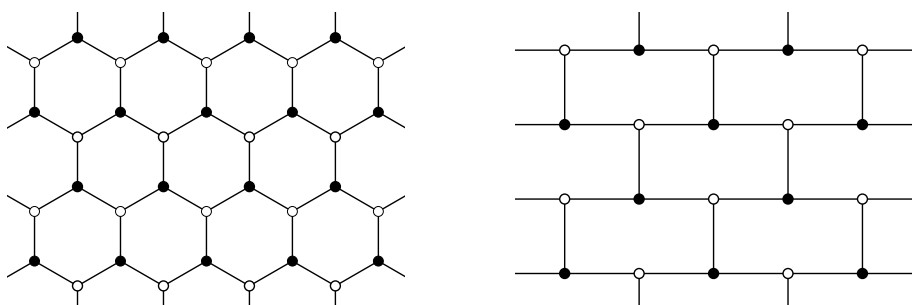

Figure 17: Honeycomb and topologically equivalent brick-wall lattice.

eliminate divergences in the derivative of the SVD with degenerate spectrum has been proposed in Ref. [66]. Here, for the case of a rotationally invariant CTMRG, the divergent term is canceled out by a particular gauge fixing of the environment tensors.

# 3 Extension to other lattices

The directional CTMRG routine on the square lattice is very convenient for its orthogonal lattice vectors and definition of the effective environments. It is therefore natural to exploit the implemented routines for different kind of lattices that can be mapped back to the square lattice. This can typically be achieved by a suitable coarse-graining, in which a collection of lattice sites on the original lattice is mapped into an effective site on the square lattice. Energy expectation values can then be directly evaluated in the coarse-grained picture as well. This is even advantageous for the AD optimization procedure, since the energy can often be computed with a smaller number of individual terms. In this section we will present the mapping for four types of lattices frequently found in condensed matter systems – the honeycomb, kagome, square-kagome and triangular lattice. Naturally, the framework can be extended by other suitable two-dimensional lattices, such as dice, square-octagon, maple-leaf and others. As an alternative to the coarse-graining approach, CTMRG methods that directly operate on the original lattice structures can also be defined [46, 105, 106].

## 3.1 Honeycomb lattice

The honeycomb, hexagonal or brick-wall lattice is of broad interest in material science and often appears in the context of quantum many-body systems. For instance, the *Kitaev honeycomb model* is a paradigmatic example hosting different kinds of phases supporting different types of anyons, both Abelian and non-Abelian [107]. We will now describe the general technical framework to simulate honeycomb lattices with the backbone CTMRG procedure described in Sec. 2.2. To this end we consider an elementary unit cell of the honeycomb lattice. Here we choose to define it along so-called $x$-links for reasons that become clear soon. Alternatively and equivalently, it could as well be defined along $y$- or $z$-links. As an example with eight different tensors on the honeycomb lattice, corresponding to four elementary unit cells is shown in Fig. 18. Coarse-graining the two lattice sites along $x$-links of the honeycomb lattice directly results in a square lattice, as shown in Fig. 19. Here, the (mapped) unit cell has size $(L_x, L_y) = (2, 2)$ with an arrangement as in Eq. (3) and Fig. 1. The green color is used to highlight the coarse-graining along $x$-links. In contrast to the regular square lattice, each coarse-grained tensor has two physical indices that can be reshaped to a single, combined index before feeding it into the CTMRG procedure. A trivial unit cell on the square lattice, consisting of only a single-site tensor, results in two different tensors on the honeycomb lattice.

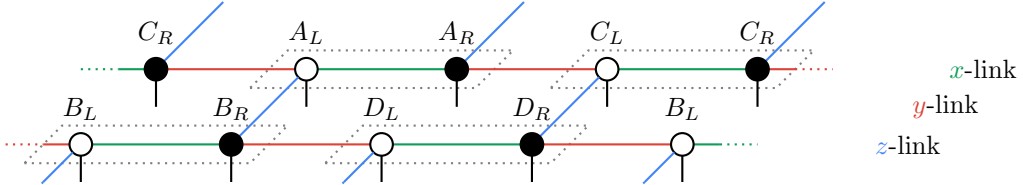

Figure 18: iPEPS Ansatz on the honeycomb lattice with four elementary unit cells, resulting in eight different lattice sites. $x$-, $y$- and $z$-links denote the three types of inequivalent links in the lattice. Coarse-graining this state to a square lattice results in a $(L_x, L_y) = (2, 2)$ configuration, with an arrangement as in Eq. (3) / Fig. 1.

The CTMRG routine can then be run as described above, just with a larger physical dimension. This does not change anything in the contractions, it is just computationally more expensive. Expectation values can now be evaluated accurately using the CTMRG environment tensors. Assuming nearest-neighbour terms again, expectation values along $x$-links can be computed by a single-site TN, while $y$- and $z$-bonds remain two-site TNs similarly to Fig. 11.

## 3.2 Kagome lattice

Another important and often encountered lattice in condensed matter physics is the kagome lattice. It is of special interest due to its corner-sharing triangles, which lead a strong geometric frustration for anti-ferromagnetic models. Using a simple mapping of the kagome lattice to a square lattice, we can directly incorporate it into our variational PEPS library. The kagome lattice is shown in Fig. 20a. Naturally, we can define a unit cell of tensors that is repeated periodically over the whole two-dimensional lattice. In our setting we consider an upward triangle on the kagome lattice as an elementary unit cell, highlighted by the gray dotted area in Fig. 20a. By choosing a coarse-graining, we can represent the three lattice sites in the unit cell by a single iPEPS tensor, which connects to its neighbours by four virtual indices. This direct mapping is shown in Fig. 20b. Nearest-neighbour links in the kagome lattice get mapped to nearest-neighbour or second-nearest-neighbour links in the square lattice. Every iPEPS site on the square lattice has a physical dimension of $p^3$. As an alternative mapping, which results in the same coarse-grained TN structure, we move from the kagome lattice to its dual, the honeycomb lattice. Here the spins live on the links instead of the vertices. The honeycomb mapping presented in Sec. 3.1 is therefore not directly applicable and additional simplex tensors are necessary to connect the lattices sites. This TN structure is shown in Fig. 21, which is commonly known as the infinite *projected entangled simplex state* (iPESS) [108]. Due to this particular mapping, three kagome lattice sites (along with two simplex tensors) are coarse-grained into

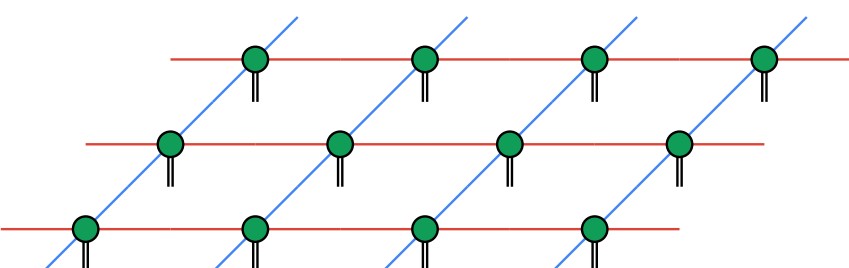

Figure 19: Using a mapping the brick-wall lattice is transformed to the square lattice. The green color of the tensors is just to highlight the coarse-graining along $x$-links, while $y$- and $z$- links remain in the network.

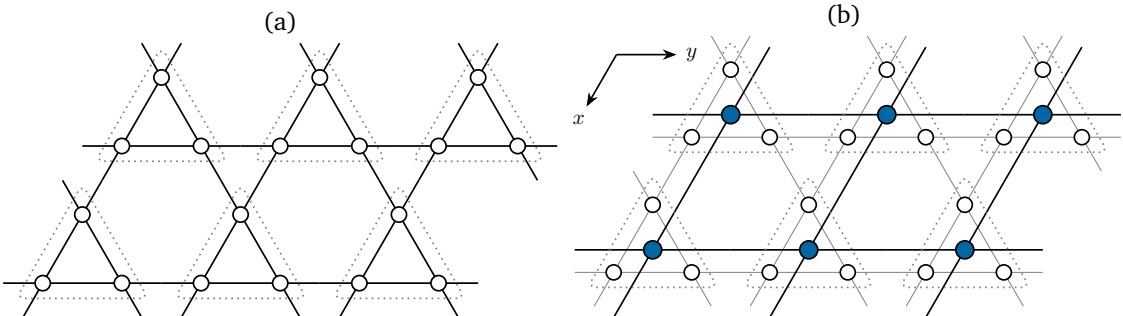

Figure 20: (a) Regular kagome lattice with corner-sharing triangles and an elementary unit cell consisting of three lattice sites. (b) Regular kagome lattice mapped to a square lattice by coarse-graining of the three spins in each unit cell.

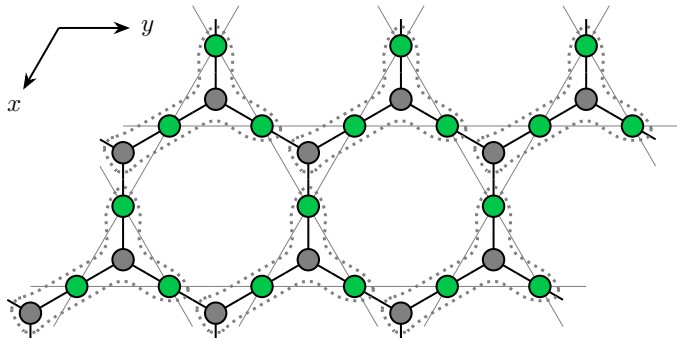

Figure 21: Honeycomb lattice (dual to the kagome lattice) with spins residing on the lattice links and additional simplex tensors on the lattice sites. Unit cells are highlighted by the gray dotted areas. Upon coarse-graining of the unit cells, the dual honeycomb lattice is mapped to the regular square lattice. Physical indices of the corresponding TN states are not shown.

a single iPEPS site on the square lattice. While the mappings in Fig. 20b and Fig. 21 result in the same square lattice TN, they differ in the number of variational parameters in the Ansatz. In the direct iPEPS Ansatz, every unit cell tensor has $p^3\chi_B^4$ parameters, while there are only $(3p\chi_B^2 + 2\chi_B^3)$ parameters for the iPESS Ansatz. Moreover, quantum correlations between lattice sites are exactly captured within the coarse-grained cluster for the iPEPS, whereas they are limited by the bulk bond dimension for the iPESS. In the ladder case, however, there is no bias between lattice sites within one cluster and sites belonging to different clusters. The nearest neighbor interactions on the kagome lattice are mapped to on-site, nearest neighbor and next-nearest neighbor interactions on the square lattice. As a concrete mapping example which has particular use in the study of the regular Heisenberg model in a magnetic field, we consider the iPEPS configuration

$$\mathcal{L} = \begin{pmatrix} A & B & C \\ B & C & A \\ C & A & B \end{pmatrix}, \tag{19}$$

on the square lattice. This configuration results in the kagome lattice structure shown in Fig. 22.



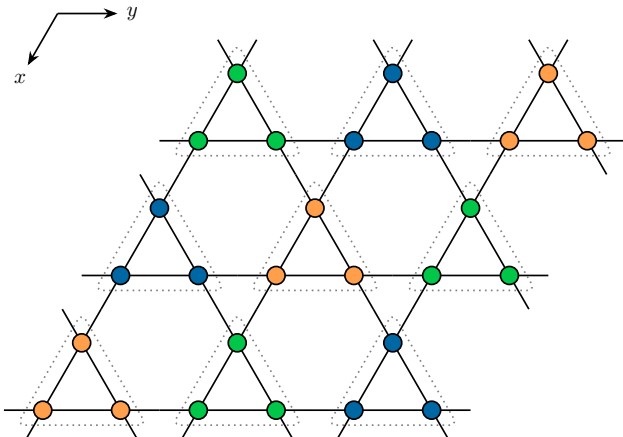

Figure 22: Kagome lattice structure corresponding to a square lattice unit cell according to Eq. (19).

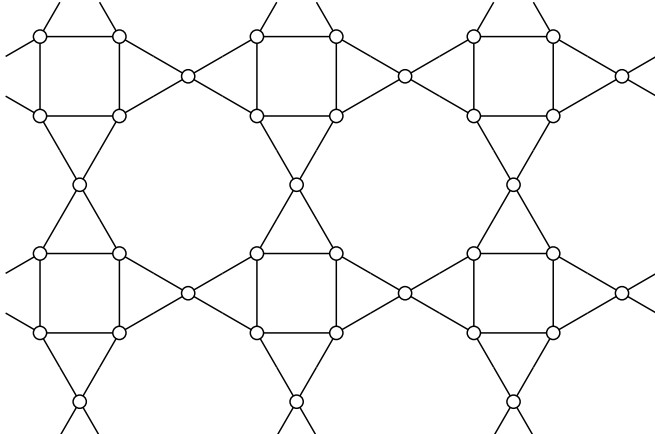

Figure 23: Square-kagome lattice. Similarly to the regular kagome lattice, it features corner-sharing triangles. The elementary unit cell consists of six sites, as shown in Fig. 24.

## 3.3 Square-kagome lattice

As a third lattice that has gained a lot of interest in recent time is the square-kagome lattice. Similar to the regular kagome lattice it features corner-sharing triangles and it is expected to host exotic quantum phases due to the geometric frustration for antiferromagnetic spin models. The square-kagome lattice structure is shown in Fig. 23. Naturally, a coarse-graining of the six spins in the elementary unit cell can be used, which directly maps the square-kagome lattice to a square lattice as depicted in Fig. 24. Following the same construction as for the regular kagome lattice, we can generalize the iPESS Ansatz to the dual of the square-kagome lattice, the so-called $(4, 8^2)$ Archimedean lattice. This results in an Ansatz with four simplex tensors and six lattice site tensors per elementary unit cell, as illustrated in Fig. 25. Counting the number of variational parameters in both TN ansätze, we find a drastic reduction in the iPESS Ansatz, again. Here the iPEPS has $p^6\chi_B^4$ parameters, while the iPESS only has $(6p\chi_B^2 + 4\chi_B^3)$ parameters for each tensor in the unit cell. In Table 1, we reinforce the difference for usual iPEPS bond dimensions, which has a strong influence on the expressivity and optimization of the different TN structures. As in the case of the kagome lattice, the first coarse-graining captures quantum correlations within the cluster exactly. While this is not the case for the iPESS mapping, it does not introduce a bias for the different lattice sites within and across

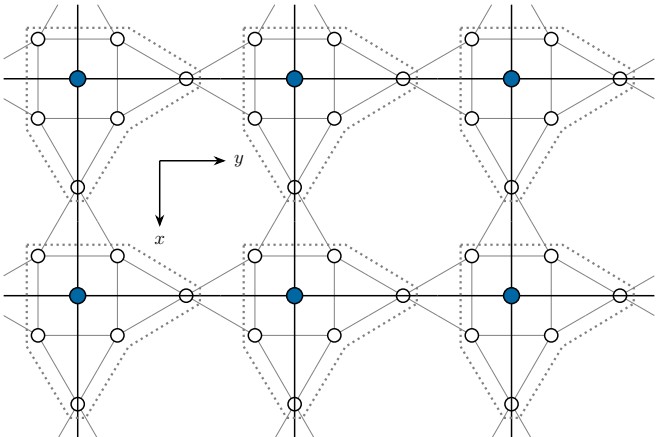

Figure 24: Regular square-kagome lattice mapped to a square lattice by coarse-graining the six spins in each elementary unit cell.

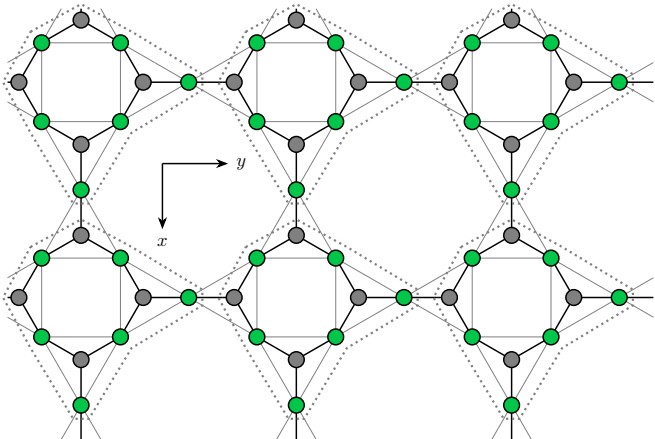

Figure 25: Square-octagon lattice (dual to the square-kagome lattice) with spins residing on the lattice links and additional simplex tensors on the lattice sites. Unit cells are highlighted by the gray dotted areas. Upon coarse-graining of the unit cells, the square-octagon lattice is mapped to the regular square lattice. Physical indices of the corresponding TN states are not shown.

clusters. Both mappings result in a large physical bond dimension of $p^6$, with $p$ the Hilbert space dimension of the original degrees of freedom (e.g., $p = 2$ for a spin-1/2). This makes especially the CTMRG routine computationally expensive. As an example we consider a two-site checkerboard pattern $((L_x, L_y) = (2, 2)$ with only two different tensors) on the square lattice, given by

$$\mathcal{L} = \begin{pmatrix} A & B \\ B & A \end{pmatrix}. \tag{20}$$

This results in a square-kagome state with twelve different lattice sites, as shown in Fig. 26.

Assuming nearest-neighbour interactions in the Hamiltonian, the ground state energy can be computed by single-site as well as horizontal and vertical two-site expectation values.

## 3.4 Triangular lattice

The triangular lattice, shown in Fig. 27 is another two-dimensional lattice variant that appears frequently in condensed matter systems. Due to its large connectivity to six nearest neighbours,

Table 1: Number of variational parameters (per elementary unit cell) in the iPEPS and iPESS TN Ansatz of the square-kagome lattice for $p = 2$, assuming real tensor elements.

| $\chi_B$ | $p^6 \chi_B^4$ | $(6p\chi_B^2 + 4\chi_B^3)$ | ratio |
|---|---|---|---|
| 2 | 1024 | 80 | 12.8 |
| 3 | 5184 | 216 | 25.0 |
| 4 | 16384 | 448 | 36.6 |
| 5 | 40000 | 800 | 50.0 |
| 6 | 82944 | 1296 | 64.0 |
| 7 | 153664 | 1960 | 78.4 |
| 8 | 262144 | 2816 | 93.1 |

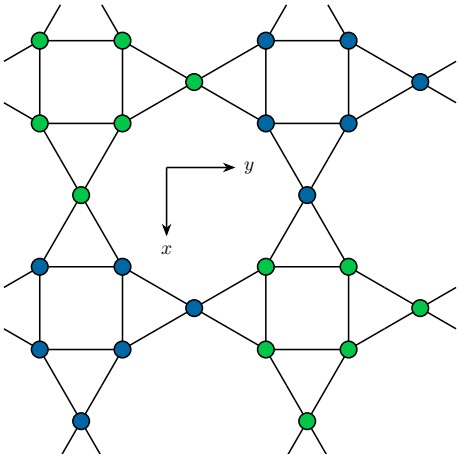

Figure 26: Square-kagome lattice structure for a square lattice unit cell according to Eq. (20). The Ansatz has twelve different lattice sites with two-site translation invariance in both $x$- and $y$-direction.

it is a typical playground for frustrated systems, hosting a variety of different quantum phases. As a consequence of this, the large connectivity makes it more challenging for numerical simulations. The triangular lattice can be directly interpreted as a square lattice with additional diagonal interactions. The entanglement between diagonal sites is then mediated by the regular virtual links in the square lattice tensor network. Nearest-neighbour interactions on the triangular lattice are again mapped to nearest-neighbour and next-to-nearest-neighbour interaction on the coarse-grained square lattice.

An alternative TN representation of the triangular lattice can be constructed using again the iPESS Ansatz. In contrast to the iPESS for kagome and square-kagome lattices, here the lattice sites have three virtual indices, too. The mapping is visualized in Fig. 28 with the iPESS Ansatz being a honeycomb lattice. Similarly to the first interpretation, this iPESS honeycomb Ansatz can be mapped to a regular square lattice with additional next-to-nearest-neighbour interactions. While the first approach as $p\chi_B^4$ parameters per unit cell tensor, the iPESS mapping only has $(p\chi_B^3 + \chi_B^3)$ coefficients. Finally, and as an alternative to the previous mappings, a reverse transformation could be used, which involves a fine-graining of the lattice sites [109].

## 3.5 Comments about different structures

In general there is no unique way to map a given lattice structure to the square lattice. The different approaches mainly differ in the number of variational parameters. While the energy

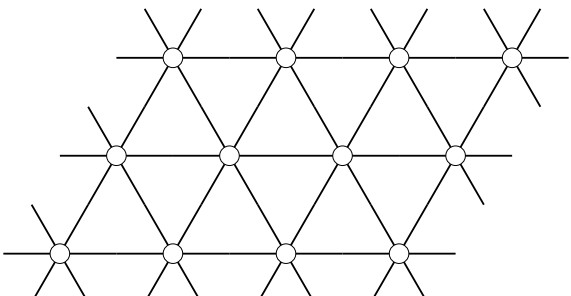

Figure 27: Regular triangular lattice with a connectivity of six, i.e., every lattice site is connected to six nearest neighbours.

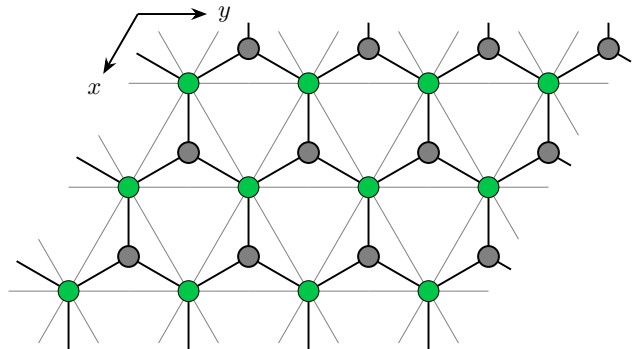

Figure 28: iPESS Ansatz for the triangular lattice consisting of only two tensors per triangular lattice site. When one lattice site and one simplex tensor are combined, the triangular lattice is directly mapped onto a regular square lattice.

for an Ansatz with fewer parameters can be optimized with fewer resources, an Ansatz with a higher variational freedom might be able to capture the physical system more accurately. At the same time the optimization becomes more complex due to the need to calculate bigger gradients. In practice, choosing the right Ansatz depends on the spatial structures of the quantum state, the amount of entanglement present in the system and the required accuracy. One strategy that works well is a step-wise optimization. In the first step one can choose, e.g., an iPESS Ansatz with fewer variational parameters. Once an optimized wave function has been found, the iPESS Ansatz is coarse-grained into a TN with a higher number of variational parameters, e.g., a direct iPEPS Ansatz. A second optimization of this more expressive Ansatz might then result in lower ground state energies. In the following sections we will present benchmarks, where several of the lowest data points have been obtained with such a two-step procedure.

## 4 Benchmarks and discussions

In this section, we will present benchmarks for a challenging and paradigmatic models on the different currently supported lattices. Due to its prominence and availability of benchmarks to different numerical techniques, we generally focus on the spin-1/2 Heisenberg antiferromagnet. The Heisenberg Hamiltonian is given by

$$H = J \sum_{\langle i,j \rangle} \vec{S}_i \cdot \vec{S}_j \,, \tag{21}$$

where $\langle i, j \rangle$ denotes nearest neighbours and $\vec{S}_i$ are the spin-1/2 operators on the lattice sites. We consider isotropic anti-ferromagnetic interactions at $J = 1.0$ throughout the benchmark section. Variational energies obtained with our implementation are denoted by *"variational update"* (VU). Where applicable, we include different TN variants (e.g., iPESS and iPEPS) in the numerical benchmarks, to highlight the effect of different numbers of variational parameters. Imaginary time-evolution in the form of a *"simple update"* (SU) on the different lattice structures can provide initial states for the variational update as discussed in Sec. 2.8.3. Whenever we use initial tensors from the SU, we add a small amount of random noise to the input tensors prior to the variational update, in order to circumvent possible local minima in the imaginary time evolution.

In the plots of this section we include the energies calculated by the mean-field environment (MF) used in the simple update. Using this approximation much larger iPEPS bond dimensions are computationally feasable but we would like to point out that this method is not guaranteed to be variational in the sense that the energy is an upper bound to the ground state energy. Thus, it is only sensible to rigorously compare results for which energy expectation values are computed by CTMRG. We include the non-variational MF energies for higher iPEPS bond dimensions for a rough comparison.

We add for each benchmark a table with the comparison of the results obtained by the simple update simulations and the best result throughout all variational updates for a fixed iPEPS bond dimension $\chi_B$. Both expectation values have been calculated by CTMRG.

## 4.1 Comments on lower bounds in variational principles

As a further conceptual point, it is important to stress that variational principles can be benchmarked as well by resorting to lower bounds to ground state energies. Such lower bounds can be efficiently computed and hold in the thermodynamic limit up to a small constant error in the energy density [110]. If the Hamiltonian $H$ is seen as being written as a sum of terms

$$H = \sum_j h_j \,, \tag{22}$$

where each $h_j$ is a patch that contains as many unit cells that can be accommodated in an exact diagonalization, then

$$\frac{\langle \psi | H | \psi \rangle}{\langle \psi | \psi \rangle} \geq E_0 \,, \quad \forall \, |\psi\rangle \,, \qquad E_0 \geq \lambda_{\min}(h_j) \,, \tag{23}$$

where $\lambda_{\min}(h_j)$ denotes the smallest eigenvalue of the patch $h_j$ with open boundary conditions. In this way, the quality of the variational principle giving rise to upper bounds to the ground state energy can be certified by lower bounds.

## 4.2 Honeycomb lattice

For the simulations of the Heisenberg on the honeycomb lattice we choose a single-site unit cell, consisting of only two different tensors on the honeycomb lattice. A mapping to the square lattice yields a fully translationally invariant iPEPS with a local Hilbert space dimension of $p^2 = 4$. We optimize the ground states on both TN structures with $2p\chi_B^3$ and $p^2\chi_B^4$ numbers of variational parameters, respectively (assuming real tensor coefficients). The model is known to be in a gapless Néel ordered phase [113–115]. Therefore, high environment bond dimensions $\chi_E$ are required to capture the large correlation lengths of the critical state. Ground state energies are reported in Fig. 29. The critical property of the ground state is already nice reflected in the significant difference between simple update MF and CTMRG expectation values. The CTMRG

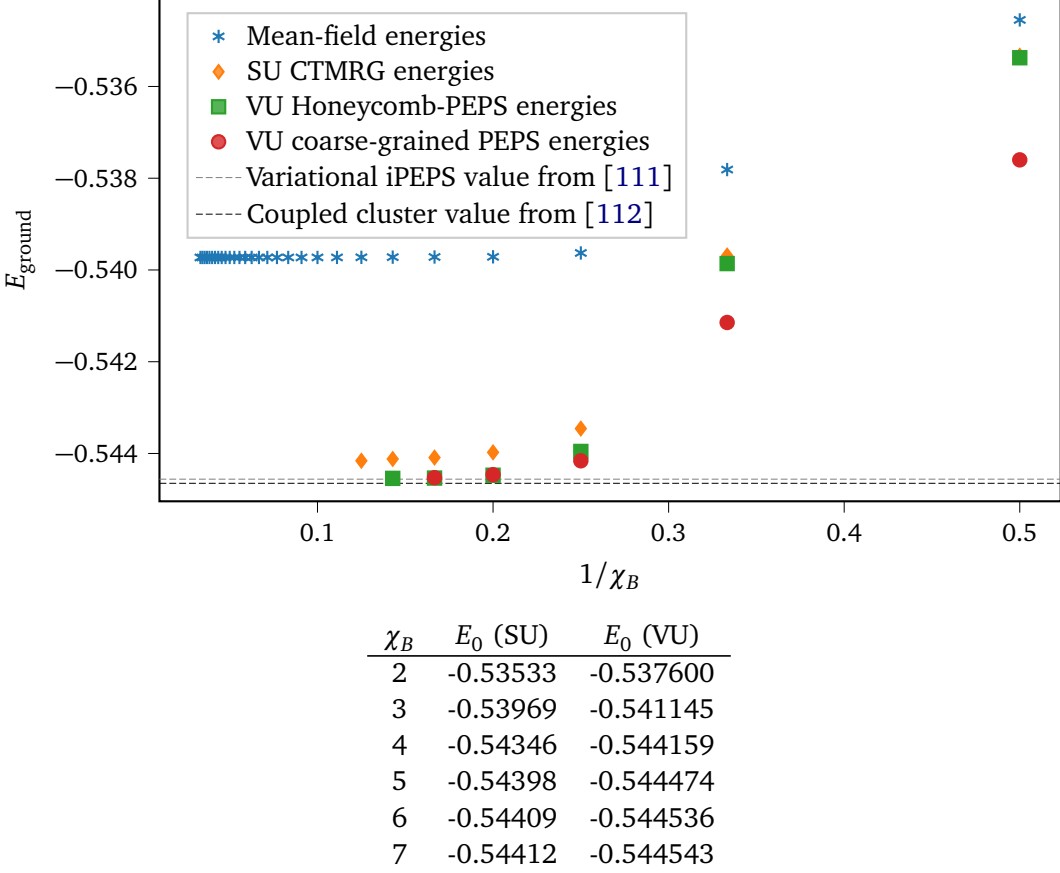

| $\chi_B$ | $E_0$ (SU) | $E_0$ (VU) |
|---|---|---|
| 2 | -0.53533 | -0.537600 |
| 3 | -0.53969 | -0.541145 |
| 4 | -0.54346 | -0.544159 |
| 5 | -0.54398 | -0.544474 |
| 6 | -0.54409 | -0.544536 |
| 7 | -0.54412 | -0.544543 |

Figure 29: Benchmarking results for the isotropic spin-1/2 Heisenberg model on the honeycomb lattice. For comparison we include the variational result obtained by an iPEPS study in Ref. [111]. Additionally, the result calculated by the coupled cluster method in Ref. [112] is shown, which is due to extrapolation not variational either.

environments treat quantum correlations much more carefully, which leads to improved energies for the infinite TN state. The VU provides lower energies than the SU with CTMRG and our results using the VU are compatible with previous results using variational iPEPS with a different CTMRG procedure [111] as well as extrapolated and thus non-variational results from the coupled cluster method [112].

## 4.3 Kagome lattice

The Heisenberg model on the kagome lattice can be considered one of the most enigmatic and well studied models in the field of frustrated magnetism [117]. While a spin liquid ground state is well established, the actual type of ground state is still under debate with different methods supporting different states (e.g., $\mathbb{Z}_2$ gapped spin liquid [118, 119], $U(1)$ gapless spin liquid [116, 120]).

Since the ground state is known to be a spin liquid state, that does not form any magnetic ordering down to zero temperature while preserving lattice translation and rotation symmetry, we use the smallest unit cells of only three sites in our simulations. The SU then works on the three-site iPESS Ansatz. The VU is performed both on the honeycomb iPESS and on a coarse-grained, fully translationally invariant iPEPS state. The number of variational parameters are hence $(3p\chi_B^2 + 2\chi_B^3)$ for the iPESS and $p^3\chi_B^4$ for the iPEPS. Again, the iPEPS state is more expressive and produces lower variational energies, that follow a smoother convergence with

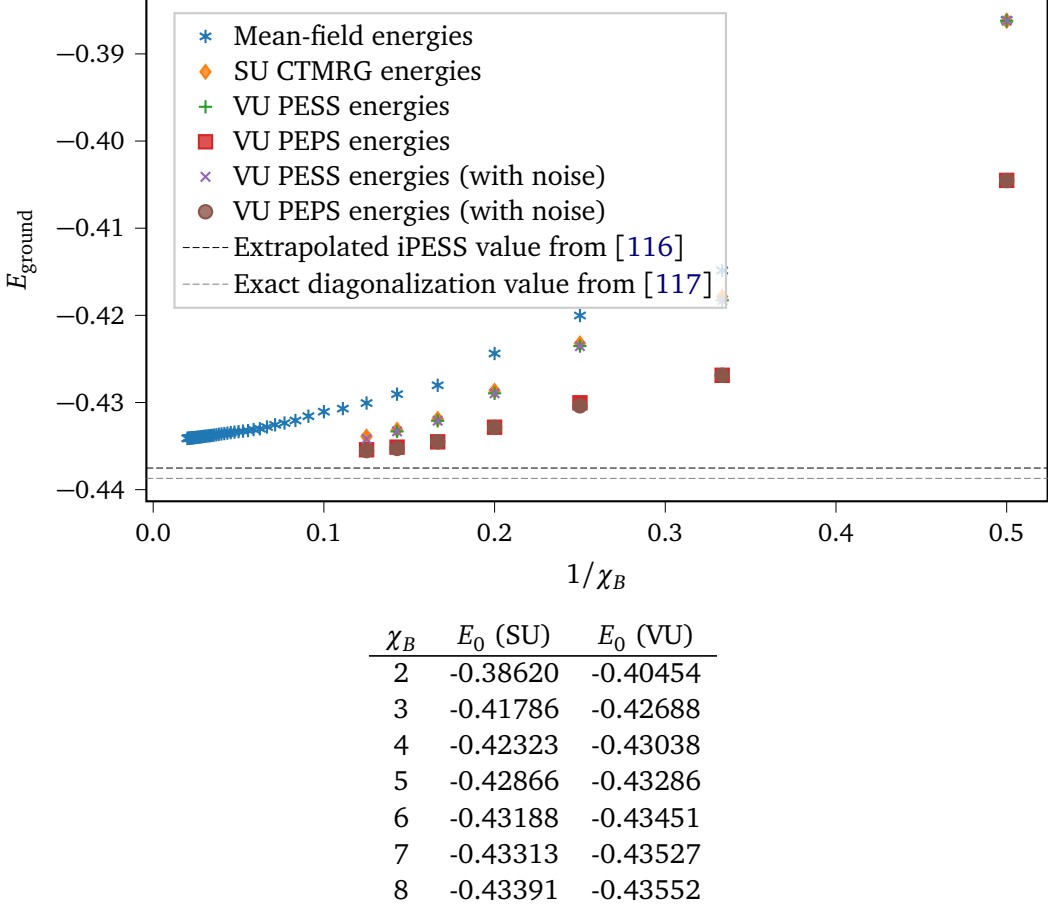

| $\chi_B$ | $E_0$ (SU) | $E_0$ (VU) |
|---|---|---|
| 2 | -0.38620 | -0.40454 |
| 3 | -0.41786 | -0.42688 |
| 4 | -0.42323 | -0.43038 |
| 5 | -0.42866 | -0.43286 |
| 6 | -0.43188 | -0.43451 |
| 7 | -0.43313 | -0.43527 |
| 8 | -0.43391 | -0.43552 |

Figure 30: Benchmarking results for the isotropic spin-1/2 Heisenberg model on the kagome lattice. For comparision, we show the outcome obtained by extrapolated iPESS results in Ref. [116], which, to be strict, is not variational as the authors noted. Additionally, we include the result computed by exact diagionalization in Ref. [117].

bond dimension $\chi_B$, see Fig. 30. The ED energy provides a lower-bound for the energy, as argued in Sec. 4.1. Our energies are compatible with other state-of-the-art numerical methods as the extrapolated iPESS result from Ref. [116], but we would like to point out that the authors noted that their results are not variational and hence the comparison is slightly tainted. Our result showcases the purpose of variational iPEPS optimization for highly frustrated systems to obtain a real upper bound to the ground state energy.

## 4.4 Square-kagome lattice

As a third benchmark model, we simulate the Heisenberg model on the square-kagome lattice, a lattice that has gained attention as a class of promising quantum spin liquid materials [123]. It consists of corner-sharing triangles, that generate a high geometric frustration similar to the kagome lattice. Its ground state has been found to be non-magnetic, however the existing subtle competition between different types of *valence bond crystal* (VBC) states has only been resolved recently in a TN study [122], in favor of a VBC with loop-six resonances. Simulations of the model are performed for a twelve-site checkerboard unit cell, as shown in Fig. 26. Results for the ground state energy are presented in Fig. 31. Due to the VBC ground state with a small correlation length and an energy gap in the model, the simple update MF and CTMRG energies are nearly identical. The variational update is performed on a so-called semi-

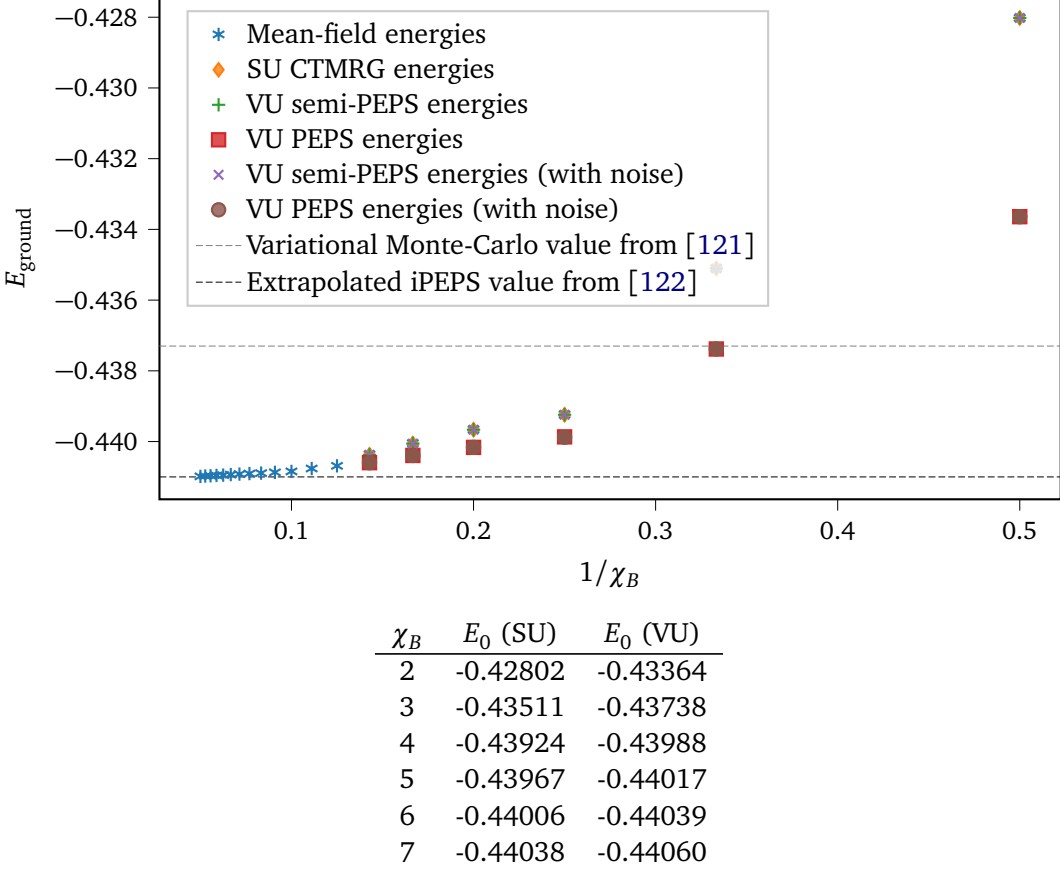

| $\chi_B$ | $E_0$ (SU) | $E_0$ (VU) |
|---|---|---|
| 2 | -0.42802 | -0.43364 |
| 3 | -0.43511 | -0.43738 |
| 4 | -0.43924 | -0.43988 |
| 5 | -0.43967 | -0.44017 |
| 6 | -0.44006 | -0.44039 |
| 7 | -0.44038 | -0.44060 |

Figure 31: Benchmarking results for the isotropic spin-1/2 Heisenberg model on the square-kagome lattice. For comparison, we include the variational Monte-Carlo results presented in Ref. [121]. Additionally, we show the extrapolated iPEPS result obtained in Ref. [122], which, to be strict, is not variational. We stress that the mean-field energies also are not variational as discussed in Sec. 4.

PEPS structure as described in Ref. [122] and also on a coarse-grained iPEPS TN as introduced in Fig. 24, a structure that is unfeasible for SU simulations due to the large imaginary time evolution operators. Although the VU cannot significantly improve the ground state energy for the semi-PEPS Ansatz, the VU on the full coarse-grained iPEPS structure improves the energies at the same bond dimension $\chi_B$. This is connected to the larger expressivity of the coarse-grained structure.

Our results outperform variational Monte-Carlo simulations in Ref. [121] and are comparable to state-of-the-art iPEPS results in Ref. [122]. We emphasize that the latter result is in the extrapolation, strictly speaking, not variational so that a comparison is slightly tainted.

## 4.5 Triangular lattice

As a last benchmark model we consider the Heisenberg model on the triangular lattice. Due to its connectivity of six, the triangular lattice exhibits a large amount of geometric frustration. The ground state is believed to be a three-sublattice 120° magnetically ordered state [125, 126]. The ground state of the Heisenberg model on the triangular lattice is computed using a three-sublattice unit cell arranged in an ABC-BCA-CAB structure. The simple update data has been produced by an iPESS Ansatz with the simplices sitting in the upward triangles (see Fig. 28). The VU is performed in two steps, using the converged iPESS state as input for second coarse-grained optimization run.

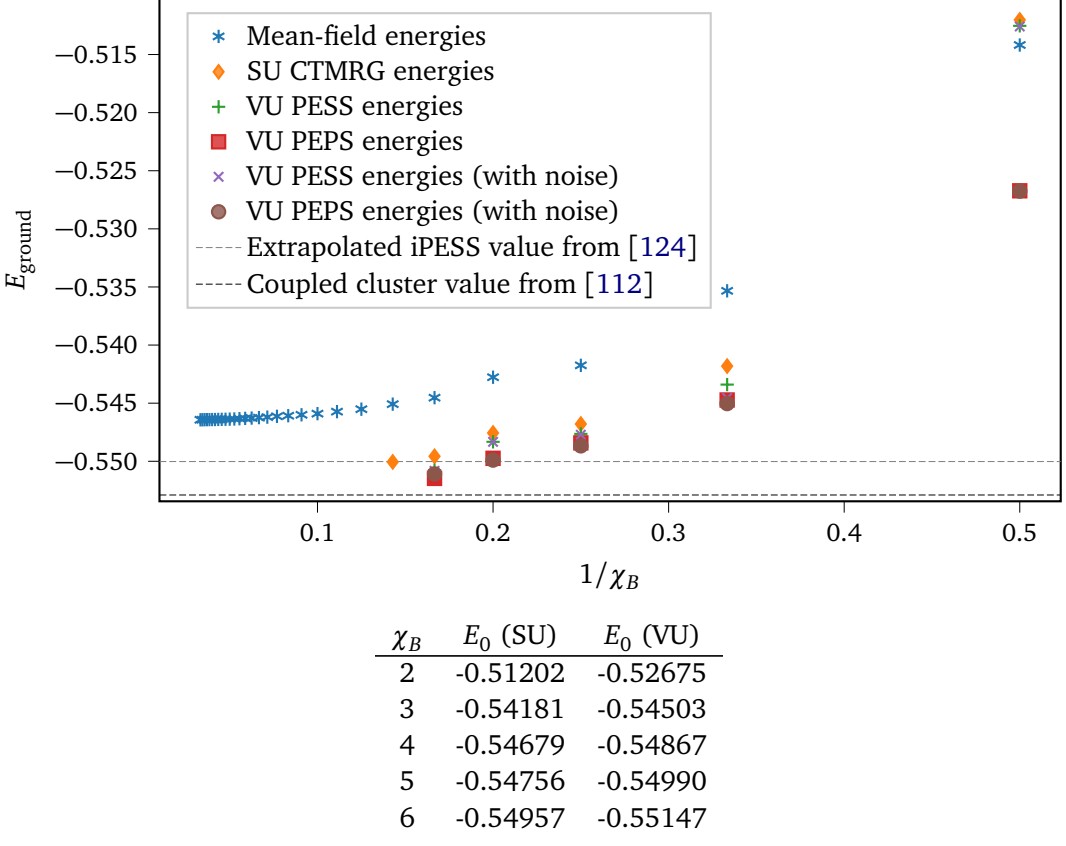

| $\chi_B$ | $E_0$ (SU) | $E_0$ (VU) |
|---|---|---|
| 2 | -0.51202 | -0.52675 |
| 3 | -0.54181 | -0.54503 |
| 4 | -0.54679 | -0.54867 |
| 5 | -0.54756 | -0.54990 |
| 6 | -0.54957 | -0.55147 |

Figure 32: Benchmarking results for the isotropic spin-1/2 Heisenberg model on the triangular lattice with an $ABC - BCA - CAB$ $3 \times 3$ unit cell structure. For comparision, we include the extrapolated, thus non-variational coupled cluster results presented in Ref. [112]. Additionally, we show the extrapolated iPESS result obtained in Ref. [124], which, to be strict, is not variational.

The results of our benchmark are shown in Fig. 32. In the case of the triangular lattice it generally helps to add some noise on the SU input state to reach better ground states and energies. We compare against a recent iPESS study based on the simple update [124], that predicts a zero-temperature magnetisation consistent with previous Monte Carlo studies [127] and additionally against a result obtained by the extrapolated, thus non-variational coupled cluster method [112]. We would like to point out that the iPESS result was extrapolated and is, strictly speaking, not variational.

## 4.6 Comments on excited states

In this work, we have primarily focused on providing a comprehensive discussion of the use of AD for the study of ground state properties of interacting quantum lattice models. It should go without saying, however, that excited states can be included in a straightforward manner. The study of excited states has first been initiated in the realm of matrix product states [128], but has later been generalized to iPEPS [129–131], allowing for constructing variational Ansatzes for elementary excitations on PEPS ground states that facilitate computing gaps, dispersion relations, and spectral weights in the thermodynamic limit.

More recently, automatic differentiation has also found its way into the optimisation of excited states [42]. The central idea is to construct the excited state with momentum $\vec{k} = (k_x, k_y)$ as a superposition of the ground state vector, perturbed by a single tensor $B$ at position

$\vec{x} = (x, y)$ and appropriate phase factors according to

$$|\phi(B)_{\vec{k}}\rangle = \sum_{\vec{x}} e^{i\vec{k}\vec{x}} |\phi(B)_{\vec{x}}\rangle \,. \tag{24}$$

The coefficients of tensor $B$ are then determined by energy minimisation of the excited state, for which AD can again be used [42,132]. In contrast to the regular ground state optimisation, here the CTMRG routine must be extended to include the appropriate phase factors in the directional absorption. Moreover, instead of only eight environment tensors per iPEPS tensor in the unit cell, the action of $B$, $B^\dagger$ and the product of $B$ and $B^\dagger$ has to be tracked in three additional sets of eight tensors.

The excited state approach can be directly extended to different lattice geometries. To this end, we have to generalize the absorption of iPEPS tensors (growing the CTMRG transfer tensors $T_1$, $T_2$, $T_3$ and $T_4$) to include the basis of the lattice, respecting relative phase factors of the basis vectors. Depending on the actual structure of the basis, a separate tensor $B_n$ is chosen as a perturbation for each of the basis site. Our implementation already contains the main building blocks of a robust and flexible CTMRG routine, calculation of gradients using AD at the fixed-point and minimisation of an energy cost function. The extension of the framework to include excited states is therefore natural. It is planned as a future feature.

## 4.7 Comments on fermionic systems

As a final comment we stress that for clarity and to be concise, we have focused in our presentation on quantum spin models. It should be clear, however, that the machinery developed here readily carries over to the study of *interacting fermionic systems*, with little modifications. Naively, one might think that the simulation of two-dimensional fermionic models is marred by substantial overheads that emerge when invoking a spin-to-fermion mapping. This is, however, not the case, and the respective book-keeping of the signs can be done with negligible overhead [133, 134]. On the formal level, such tensor networks involve a particular choice of what is called a spin structure [135, 136]. Practically speaking, one can modify much of the bosonic code for PEPS to the fermionic setting, readily incorporating the relevant signs to capture interacting fermions, in what is called *fermionic PEPS* [133, 137, 138]. This insight is important as some of the most compelling test cases of interacting quantum many-body systems are of a fermionic nature.

## 5 Conclusion and prospects

In this review we present a comprehensive introduction into automatic differentiation in the context of two-dimensional tensor networks, leading to the recently emerging variational iPEPS framework for ground state optimization. We provide implementation details and discuss obstacles that arise in practice, as well as techniques to mitigate these. At the same time, we coherently present ideas that have to date only been mentioned in a fragmented fashion in the literature. We hope that the present work can serve as a useful reference and review in the variational study of 2d tensor networks.

This work accompanies the variational iPEPS library *variPEPS*, a comprehensive and versatile code base for optimizing iPEPS in a general setting. We expect this library to be a helpful tool for performing state-of-the-art tensor network analyses for a wide range of physical models, featuring multiple two-dimensional lattices.

The library is designed to be extended with additional simulation techniques based on automatic differentiation, such as excited states and structure factors.

Table 2: Summary of the estimated lower bound of the carbon cost generated during the development of this work. The estimations have been calculated using the examples of the Scientific $CO_2$nduct project [139] and include the costs of the numerical calculations and air travel for collaborations.

| Numerical simulations | |
|---|---|
| Total Kernel Hours [h] | $\geq 255276$ |
| Thermal Design Power Per Kernel [W] | 12 |
| Total Energy Consumption Simulations [kWh] | $\geq 3063$ |
| Average Emission Of $CO_2$ In Germany [kg/kWh] | 0.441 |
| Total $CO_2$-Emission For Numerical Simulations [kg] | $\geq 1351$ |
| Were The Emissions Offset? | **Yes** |
| **Air Travel** | |
| Total $CO_2$-Emission For Air Travel [kg] | 924 |
| Were The Emissions Offset? | **Yes** |
| Total $CO_2$-Emission [kg] | $\geq 2275$ |

The *variPEPS* library is publicly available in both a Julia and a Python version on GitHub [56], with stable references in the corresponding Zenodo repositories [57, 58].

## 5.1 $CO_2$-emissions table

For the sake of completeness and for promoting carbon footprint awareness, we display an estimated lower bound of the carbon emissions generated during the course of this work in Table 2.

# Acknowledgments

We acknowledge inspiring discussions with Ji-Yao Chen, Andreas Haller, Juraj Hasik, Augustine Kshetrimayum, Alexander Nietner and Niklas Tausendpfund. We would like to particularly thank Boris Ponsioen, who has for shared valuable insights, and Frederik Wilde, who has helped us to get the details of the automatic differentiation and the custom fixed-point derivative correct. We would like to thank the ZEDV (IT support) of the physics department, Freie Universität Berlin, for computing time and their technical support, particularly we thank Jörg Behrmann and Jens Dreger.

For the Python version we thank the developers of the JAX framework [140, 141] for their work to provide an AD framework with optimized numerical operations and their technical support during the development of this work. We also acknowledge the use of the TensorKit package [91] in the Julia version of the code and wish to advertise the open source libraries of the Quantum Ghent group in this context [142]. We make use of the Zygote [143] package for AD in the Julia programming language.

The work has been discussed and refined during the workshop "Tensor Networks: Mathematical Structures and Novel Algorithms (2022)" at the Erwin Schrödinger International Institute for Mathematics and Physics in Vienna and the workshop "Entanglement in Strongly Correlated Systems (2023)" at the Centro de Ciencias de Benasque Pedro Pascual. We thank the organizers for their hospitality and work.

**Funding information**  E. L. W. thanks the Studienstiftung des deutschen Volkes for support. This work has been funded by the Deutsche Forschungsgemeinschaft (DFG, German Research Foundation) under the project number 277101999 – CRC 183 (project B01), for which this constitutes an inter-node publication involving both Cologne and Berlin, and the BMBF (MUNIQC-Atoms, FermiQP). It has also received funding from the Cluster of Excellence MATH+ and from the Quantum Flagship (PasQuans2). The authors gratefully acknowledge the Gauss Centre for Supercomputing e.V. (www.gauss-centre.eu) for funding this project by providing computing time through the John von Neumann Institute for Computing (NIC) on the GCS Supercomputer JUWELS at the Jülich Supercomputing Centre (JSC) (Grant NeTeNeSyQuMa) and the FZ Jülich for JURECA (institute project PGI-8) [144].

## A  Adjoint functions and variables

In the literature it is common to use so called *adjoint functions* and *adjoint variables* when using backwards-mode AD. These adjoint functions map the adjoint variables onto each other, as in Eq. (15) when building up the gradient. In this section, we will briefly introduce the basic notation of adjoint functions and variables following Ref. [145]. Explicit constructions of adjoint functions, which are vector-Jacobian-products in the practical implementation, for a large number of useful operations including those for the iPEPS use-case can be found in Refs. [145–147].

As an example throughout this section, we consider the function $h$, composed out of two primitive functions $h_1$ and $h_2$ which are concatenated as

$$
\begin{aligned}
h &= h_2 \circ h_1\,, \\
h_1 &: M_{n \times n} \times M_{n \times n} \to M_{n \times n}\,, \\
h_2 &: M_{n \times n} \to \mathbb{R}\,,
\end{aligned}
\tag{A.1}
$$

with variables $(A, B) \in M_{n \times n} \times M_{n \times n}$, $C \in M_{n \times n}$ and $x \in \mathbb{R}$. We start by examining the differential of the output variable $x$

$$
dx = \frac{\partial h_2}{\partial C} dC =: \sum_{i,j} \bar{C}_{i,j} dC_{i,j} = \mathrm{Tr}(\bar{C}^{\mathsf{T}} dC)\,.
\tag{A.2}
$$

In the first equation, we have suppressed the sum over the indices of $C$. Eq. (A.2) defines the adjoint variable $\bar{C}$ of $C$. We see that the adjoint variable $\bar{C}$ is the derivative of the scalar output of the function $h_2$ w.r.t. $C$. Thus, for the case of a scalar output the variable $C$ and the adjoint variable $\bar{C}$ have the same dimension. Now, in order to get the gradient $\nabla h$ we are interested in the derivative of the output w.r.t. the input variables $(A, B)$. To this end we consider the differential of the intermediate variable

$$
dC = \frac{\partial h_1}{\partial A} dA + \frac{\partial h_1}{\partial B} dB\,.
\tag{A.3}
$$

Inserting this into Eq. (A.2), we obtain

$$
dx = \mathrm{Tr}\!\left( \underbrace{\bar{C}^{\mathsf{T}} \frac{\partial h_1}{\partial A}}_{\bar{A}^{\mathsf{T}}} dA \right) + \mathrm{Tr}\!\left( \underbrace{\bar{C}^{\mathsf{T}} \frac{\partial h_1}{\partial B}}_{\bar{B}^{\mathsf{T}}} dB \right)\,.
\tag{A.4}
$$

Here we have already implicitly used the adjoint function $\bar{h}_1$ that maps the adjoint variable $\bar{C}$ to the adjoint variables $\bar{A}$ and $\bar{B}$ according to

$$
\bar{h}_1 : \bar{C}^{\mathsf{T}} \mapsto (\bar{A}^{\mathsf{T}}, \bar{B}^{\mathsf{T}})^{\mathsf{T}} = \left( \bar{C}^{\mathsf{T}} \frac{\partial h_1}{\partial A}, \bar{C}^{\mathsf{T}} \frac{\partial h_1}{\partial B} \right)^{\mathsf{T}}\,.
\tag{A.5}
$$

Given the fact that we are dealing with a scalar output variable $x$, we recall that $\bar{C}$ can be considered a vector, such that the adjoint function is a vector-Jacobian-product (vJP). We can see that the this maping of the adjoint variables with adjoint functions eventually produces the gradient

$$
\begin{aligned}
\nabla h = \left(\bar{A}, \bar{B}\right) &= \left(\frac{\partial h_1}{\partial A}\bar{C}, \frac{\partial h_1}{\partial B}\bar{C}\right) \\
&= \left(\frac{\partial h_1}{\partial A}\frac{\partial h_2}{\partial C}, \frac{\partial h_1}{\partial B}\frac{\partial h_2}{\partial C}\right).
\end{aligned}
\tag{A.6}
$$

# B Automatic differentiation for complex variables

Some extra attention has to be given to the case in which the primitive functions are complex valued. This is because not all functions one might want to consider are complex-differentiable (holomorphic) and as such the derivative depends on the direction we move in the complex plane when taking the limit for the derivative. In such a case one needs to resort to the calculus of two sets of independent real variables. For a generic function $f : \mathbb{C} \to \mathbb{C}$ this can be done by treating $x$ and $y$ in $z = x + iy$ as independent variables or alternatively, by choosing $z$ and $z^*$ and making use of Wirtinger calculus. However we should also note that in the iPEPS use case we deal with a function $E : \mathbb{C}^n \to \mathbb{R}$, which removes the necessity to think about holomorphism.

# C The implicit function theorem and its use at the CTMRG fixed-point

In this section, we are going to present an alternative approach to taking the derivative of the energy function by utilizing the fixed point of the CTMRG procedure. To this end, we can make use of the implicit function theorem [148] to calculate the derivative of the full fixed-point routine. Our discussion will follow the description of Refs. [149, 150]. Differentiating Eq. (16) on both sides we end up with

$$
\partial_A e^*(A) = \partial_A c(A, e^*) + \partial_{e*} c(A, e^*)\partial_A e^*(A).
\tag{C.1}
$$

Introducing the shorthand writing for the Jacobians $L = \partial_A c(A, e^*(A))$ and $K = \partial_{e*} c(A, e^*(A))$ and rearranging the equation we find

$$
\begin{aligned}
\partial_A e^*(A) &= (L + K\partial_A e^*(A)) \\
&= \left(\sum_{n=0}^{\infty} K\right) L = (\mathbb{1} - K)^{-1} L.
\end{aligned}
\tag{C.2}
$$

As discussed in Appendix A, we aim at finding the adjoint function of the CTMRG iteration at the fixed point, which is a *vector-Jacobian product* (vJP) $\mathbf{v}^{\mathsf{T}} \partial_A e^*(A)$. Inserting Eq. (C.2) yields

$$
\mathbf{v}^{\mathsf{T}} \partial e^*(A) = \mathbf{v}^{\mathsf{T}}(\mathbb{1} - K)^{-1} L = \mathbf{w}^{\mathsf{T}} L,
\tag{C.3}
$$

where we have introduced $\mathbf{w}^{\mathsf{T}} := \mathbf{v}^{\mathsf{T}}(\mathbb{1} - K)^{-1}$. The second equality in the equation above can be rearranged into another fixed-point equation

$$
\mathbf{w}^{\mathsf{T}} = \mathbf{v}^{\mathsf{T}} + \mathbf{w}^{\mathsf{T}} K.
\tag{C.4}
$$

Here $\mathbf{w}^\mathsf{T} K$ is another vJP but this time only dependent of the derivative of a single absorption step evaluated at the fixed-point of the CTMRG routine. Solving Eq. (C.4) we can find $\mathbf{w}^\mathsf{T}$ to calculate the vJP of the CTMRG routine from Eq. (C.3). In the end we reduced the naive effort of unrolling the fixed-point iterations to just calculate the derivative of a single CTMRG iteration and another fixed-point iteration which both are much less memory intensive.

# D Automatic differentiation in the language of differential geometry

In order to unify the different frameworks for thinking about forward- and backwards-mode AD, we will briefly introduce a mathematical notation for AD. It also serves to give some more precise meaning to the terms "push-forward" and "pullback", that are sometimes used in forward- and backwards-mode AD discussions, respectively. For this we first recall the general concept of a push-forward and a pullback for the simple case of functions and distributions. Imagine two functions $f : M \to N$ and $g : N \to \mathbb{R}$. The *pullback* of $g$ along $f$ allows us to construct a function $f^*g : M \to \mathbb{R}$ for which the domain of the function $g$ is "pulled back" to the domain of the function $f$. This is done by a simple concatenation of $f$ and $g$

$$f^*g(\underbrace{m}_{\in M}) = (g \circ f)(m) = g(f(m)). \tag{D.1}$$

This construction can now be used to define a *push-forward* on the dual objects of the functions under integration. These dual objects are distributions. With a distribution, we can integrate a function

$$\int_M \bullet \, \mu : \mathcal{F}(M) \to \mathbb{R}, \tag{D.2}$$
$$f \quad \mapsto \int_M f\mu,$$

where $\mathcal{F}(M)$ are just the functions on $M$ and $\mu$ is the distribution. Given such a distribution on $M$ we can now integrate functions on $M$. The push-forward $f_*\mu$ of $\mu$ allows us to integrate functions on $N$ by defining a distribution that is "pushed forward" to $N$. This works as

$$\int_N h(f_*\mu) =: \int_M (f^*h)\mu, \tag{D.3}$$

where $h$ is a function on $N$.

This type of construction for the pullback and push-forward generalizes to many mathematical objects that have a *pairing dual*. The relevant mathematical objects for AD are the derivative $\partial/\partial x_i$ and its pairing dual, the differential $dx_i$.

It might be useful, beyond the conceptual clarity of this notation, to look at AD in this way because one can easily imagine situations where the intermediate data of a function is restricted by constraints such that the "data-space" becomes geometrically non-trivial. An example could be vectors in $\mathbb{R}^n$ restricted to unit length or matrices in $M_{n,m}$ restricted to be unitary. We note that an optimisation in these situations requires some additional concepts, like finding a path on the given space from a tangent vector. This requires some extra care and is not discussed here.

We now introduce the mathematical notation that we need in order to talk about AD in this language. We will not be particularly rigorous in this endeavour and leave out all details that are not explicitly needed. We start with a manifold $M$ on which we can consider points

$p \in M$, as well as functions $f : M \rightarrow \mathbb{R}$. For each point $p \in M$ we can define a vector space $T_p M$ (call it the tangent-space at $p$) of tangent-vectors at that point. The elements in $T_p M$ act like derivatives on functions on $M$

$$\text{e.g.:} \quad \frac{\partial}{\partial x_i} = e_i \in T_p M \,, \quad \frac{\partial}{\partial x_i}(f) = \frac{\partial f}{\partial x_i} \,.$$

Here we have assumed that we have equipped the manifold $M$ with coordinates via a chart $\phi : M \rightarrow \mathbb{R}^m$ around the point p, where $m = \dim(M)$. Our tangent-space $T_p M$ has dimension $m$ and we can choose a canonical basis

$$\left\{ \frac{\partial}{\partial x_i}, \ldots, \frac{\partial}{\partial x_m} \right\} = \{e_1, \ldots, e_m\} \,.$$

One further defines the dual vector space $T_p^* M$ of the tangent vector space, called cotangent-space. This cotangent-space contains the dual vectors to the derivatives $\frac{\partial}{\partial x_i}$. These cotangent vectors from the cotangent-space are the differentials $dx_i$. The cotangent-space also has dimension $m$ and we can choose the canonical basis

$$\{dx_1, \ldots, dx_m\} \,.$$

Obviously, given the canonical basis for the tangent-space and cotangent-space we can expand arbitrary vectors in these spaces in the basis. Take $v \in T_p M$ and $df \in T_p^* M$ we can expand as

$$v = \sum_i v_i \frac{\partial}{\partial x_i} = \sum_i v_i e_i \,, \tag{D.4}$$

$$df = \sum_i \frac{\partial f}{\partial x_i} dx_i \,. \tag{D.5}$$

We have a pairing between the derivatives that live in the tangent-space $T_p M$ and the differentials that live in $T_p^* M$ as

$$dx_j \left( \frac{\partial}{\partial x_i} \right) := \frac{\partial x_j}{\partial x_i} = \delta_{i,j} \,. \tag{D.6}$$

Note that by this pairing relation we see that tangent and cotangent vectors are "pairing duals" and we can use an analogous construction for pullbacks and push-forwards as we did for functions and distribution above. Since $T_p M$ and $T_p^* M$ are isomorphic, we can introduce a correspondence transformation between the canonical basis of the two spaces

$$\bullet^\flat : T_p M \rightarrow T_p^* M \,, \quad e_i \mapsto dx_i = e_i^\flat \,, \tag{D.7}$$

$$\bullet^\sharp : T_p^* M \rightarrow T_p M \,, \quad dx_i \mapsto e_i = dx_i^\sharp \,. \tag{D.8}$$

We now have assembled all nessesary tools to formulate what a "gradient" is in this language. It is given by

$$\nabla f := (df)^\sharp \,, \tag{D.9}$$

which matches the common formula

$$\nabla f = \left( \sum_i \frac{\partial f}{\partial x_i} dx_i \right)^\sharp = \sum_i \frac{\partial f}{\partial x_i} e_i$$
$$= \left( \frac{\partial f}{\partial x_1}, \ldots, \frac{\partial f}{\partial x_m} \right) \,, \tag{D.10}$$

where we have taken $e_i$ just as the $i$-th unit vector of $T_p M$.

Now it is easy to construct the pullbacks and push-forwards in this context analogous to our treatment of functions and distributions. For this we start from manifolds $M$ and $N$ with points $p \in M$ and $q \in N$, and with the two functions $f : M \to N$ and $g : N \to \mathbb{R}$. We can consider a differential $dg \in T_q^*N$ which we want to "pull back" along the function $f$ and associate it with and element of $T_{f^{-1}(q)}^*M$, where $f^{-1}(q) \in M$. We do this with the familiar definition

$$\underbrace{f^*dg}_{\in T_{f^{-1}(q)}^*M} := d(g \circ f), \tag{D.11}$$

which uses a concatenation of $f$ and $g$ just as in the first example. For a tangible example consider $g = x_i$ to be a coordinate function. We then get $f^*dx_i = d(x_i \circ f) = d(f_i)$. As before the push-forward can be defined via the pullback just as we had done for functions and distributions. In this case, we start with a tangent vector $\frac{\partial}{\partial x_i}$ in $T_pM$ and want to "push it forward" along $f$ into $T_{f(p)}N$. This works as

$$\left( f_* \underbrace{\overbrace{\left( \frac{\partial}{\partial x_i} \right)}^{\in T_pM}}_{\in T_{f(p)}N} \right)(g) := \frac{\partial}{\partial x_i}(f^*g) = \frac{\partial}{\partial x_i}(g \circ f). \tag{D.12}$$

Now that we are equipped with the pullback and push-forward of differentials and derivatives we see how the gradient is calculated in the forward- and backward-mode AD. For this we will go back to our neat example from Sec. 2.4 and slightly generalize. Say, we would like to take the gradient $\nabla E$ of a function that is composed of three primitive functions $E = f_3 \circ f_2 \circ f_1$. We say these primitive functions map between manifolds

$$E : M_1 \xmapsto{f_1} M_2 \xmapsto{f_2} M_3 \xmapsto{f_3} \mathbb{R}. \tag{D.13}$$

Lets first look at what happens when we build the gradient using backwards-mode AD. In this case we start with the differential $df_3$ of the last primary function of $E$. This differential lives in $T_k^*M_3$, where $k \in M_3$ is a point in $M_3$. We can now use the pullback along the functions $f_2$ and then $f_1$ to pull back this differential to $M_1$

$$f_1^*(f_2^*(df_3)) \xleftarrow{\text{pullback}} f_2^*(df_3) \xleftarrow{\text{pullback}} df_3. \tag{D.14}$$

With the definitions above we see that in this way we construct the gradient

$$f_1^*(f_2^*(df_3)) = f_1^*((d(f_3 \circ f_2))) = d(f_3 \circ f_2 \circ f_1) = dE. \tag{D.15}$$

With our identification between tangent and cotangent vectors we finalize to $\nabla E = (dE)^\sharp$. If we express the differential that we start from $df_3$ in coordinates, we straightforwardly obtain the product of Jacobians as a result for the gradient. This also establishes the connection to the adjoint functions we talked about in the previous section and the vector-Jacobian product as discussed in Sec. 2.4.

In the case of forward-mode AD we start from a tangent vector $\frac{\partial}{\partial x_i}$, which lives in $T_lM_1$, where $l \in M_1$ is a point in $M_1$. We can now push this tangent vector forward into a tangent space of $M_3$ with successive push-forwards along $f_1$ followed by $f_2$

$$\frac{\partial}{\partial x_i} \xrightarrow{\text{push-forward}} f_{1*}\left( \frac{\partial}{\partial x_i} \right) \xrightarrow{\text{push-forward}} f_{2*}\left( f_{1*}\left( \frac{\partial}{\partial x_i} \right) \right). \tag{D.16}$$

With the definitions for the push-forward we see that the gradient we obtain in this way is given by

$$
\begin{aligned}
\sum_i f_{2*}\left(f_{1*}\left(\frac{\partial}{\partial x_i}\right)\right)(f_3)\, e_i &= \sum_i f_{1*}\left(\frac{\partial}{\partial x_i}\right)(f_3 \circ f_2)\, e_i \\
&= \sum_i \frac{\partial}{\partial x_i}\underbrace{(f_3 \circ f_2 \circ f_1)}_{=E}\, e_i \\
&= \nabla E\,.
\end{aligned}
\tag{D.17}
$$

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
