# Peer review of "An introduction to infinite projected entangled-pair state methods for variational ground state simulations using automatic differentiation"

_SciPost Physics Lecture Notes, doi:SciPost Phys. Lect. Notes 86 (2024)_

## Round 1 · Referee Report · Anonymous (Referee 1) · 2024-6-10

Strengths

  • a proper introduction with a lot of references, showing the versatility of variational PEPS algorithms for simulating ground states of strongly-correlated quantum lattice systems in two dimensions
  • a pedagogical explanation of all the necessary steps for implementing the CTMRG+AD procedure for optimizing PEPS
  • a description of possible pitfalls in applying this methodology
  • illustrative benchmarks
  • an accompanying software package, both in Python and Julia
  • the language is good and easy to follow

Weaknesses

  • alternative methods for contracting PEPS are not discussed, although these are common practice in the community and can be combined with AD in a very similar way as CTMRG-based contractions
  • the use of symmetries in PEPS representations (either internal or spatial symmetries) is not discussed, although this is crucial to obtain state-of-the-art results in many cases

Report

In this work, the authors give a pedagogical introduction to the variational optimization of infinite projected entangled pair states by combining the CTMRG algorithm with automatic differentiation (AD). These lecture notes will be very valuable for newcomers in the field, allowing them to use existing software packages with the proper insight into the inner workings of the method.

The CTMRG+AD methodology is a very good choice for optimizing PEPS representations, but it is not the only viable option. In particular, a comparison between boundary MPS and CTMRG methods shows that the methods are competitive for contracting PEPS [1]. Furthermore, boundary MPS contraction can be combined with AD to yield an efficient optimization code [2]. It is a legitimate choice for these lecture notes to focus on one method, but the authors should state explicitly that other methods are equally viable options.

A crucial aspect in obtaining PEPS results for challenging problems is the use of symmetries, both spatial and internal. Regarding the former, the authors discuss different lattice structures, mapping each instance to the square lattice. This mapping typically violates the symmetries of the original lattice, potentially leading to severe artefacts. This is not discussed. Moreover, in many works the use of spatial symmetries such as reflection or rotation have been used to constrain the number of variational parameters in the PEPS tensor. The CTMRG+AD methodology should be well suited to optimize within this constrained space.

Regarding the internal symmetries. It should be noted that there are many software packages that can optimize PEPS with internal symmetries (abelian and non-abelian), and there are many works where this is used with great success. The authors should discuss this feature.

[1] Phys. Rev. B 105, 195140 (2022)
[2] Phys. Rev. B 108, 085103 (2023)

Requested changes

1- An explicit discussion of alternative methodologies for contracting and optimizing PEPS should be included, leaving the reader to make an informed choice.

2- A discussion of the use of symmetries in PEPS, and how these can be readily integrated in the CTMRG+AD methodology

Recommendation

Ask for minor revision

  • validity: good
  • significance: high
  • originality: low
  • clarity: top
  • formatting: perfect
  • grammar: perfect

Author:  Jan Naumann  on 2024-08-20  [id 4706]

(in reply to Report 1 on 2024-06-10)

Dear Referee,

We want to thank your for your valuable comments and the detailed report on our manuscript. We have revised the draft by incorporating the suggestions made in the report by adding a paragraph discussing alternative methods for contractions as well as a new section about the extension of the framework to include physical symmetries in the ansatz.

Yours sincerely,
The authors

---

## Round 2 · Referee Report · Anonymous (Referee 1) · 2024-8-22

Report

I would like to thank the authors for making the changes that I have suggested, I think the Lecture Notes are ready for publication.

Recommendation

Publish (meets expectations and criteria for this Journal)

---

## Round 2 · Author Response

Dear Editor, Dear Referee,

We want to thank the referee for their valuable comments and the detailed report on our manuscript. We have revised the draft by incorporating the suggestions made in the report by adding a paragraph discussing alternative methods for contractions as well as a new section about the extension of the framework to include physical symmetries in the ansatz.

For your convenience, we have highlighted the changes in red colour in the resubmitted manuscript.

Yours sincerely,
The authors

---

## Round 2 · List of Changes

• In the introduction of Chapter 2, we have added a paragraph discussing alternative methods for contractions. We point the reader to the relevant literature, particularly with the focus on alternative methods incorporating a variational ansatz. (Lines 152-160).

  • We have added Section 2.7 focusing on the ability of tensor network ansätze to incorporate physical symmetries exactly. We mention the relevant methodology and point the reader to software implementations in this field. (Lines 538 - 560).

---

## Editorial Decision

published